# Inferring the ice sheet sliding law from seismic observations: A Pine Island Glacier case study

Kevin Hank[1], Robert J. Arthern[1], C. Rosie Williams[1], Alex M. Brisbourne[1], Andrew M. Smith[1], James A. Smith[1], Anna Wåhlin[2], and Sridhar Anandakrishnan[3]

[1]Natural Environment Research Council, British Antarctic Survey, High Cross, Madingley Road, Cambridge, CB3 0ET, United Kingdom
[2]Department of Marine Sciences, University of Gothenburg, Gothenburg, Sweden
[3]Department of Geoscience, Pennsylvania State University, University Park, Pennsylvania, 16802, USA
[*]kevhan@bas.ac.uk

**Correspondence:** Kevin Hank (kevhan@bas.ac.uk)

**Abstract.** The response of the Antarctic ice sheet to climate change and its contribution to sea level under different emission scenarios are subject to large uncertainties. A key uncertainty is the slipperiness at the ice sheet base and how it is parameterized in glaciological projections. Alternative formulations of the sliding law exist, but very limited access to the ice base makes it difficult to validate them. Here, the Viscous Grain-Shearing (VGS) theory of acoustic propagation in granular material, together with independent estimates of grain diameter and porosity from sediment cores, is used to relate the effective pressure, which is a key control of basal sliding, to seismic observations recovered from Pine Island Glacier, Antarctica. With basal shear stress and sliding speed derived through satellite observations of ice flow and inverse methods, the new Bayesian sliding law inference – VGS (BASLI–VGS) approach enables a comparison of basal sliding laws within a Bayesian model selection framework. The presented direct link between seismic observations and sliding law parameters can be readily applied to any acoustic impedance data collected in glacial environments underlain by granular material. For rapidly sliding tributaries of Pine Island Glacier, these calculations provide support for a Coulomb-type sliding law and widespread low effective pressures.

*Copyright statement.* TEXT

## 1 Introduction

Large uncertainties accompany sea level rise projections for the 21st century. Relative to 1900, the estimates vary between $\sim 50$ and $> 100$ cm (IPCC Core Writing Team, H. Lee and J. Romero (eds.), 2023). This uncertainty hampers the formulation of adaptation strategies. A key source of uncertainty is the slipperiness of the bed beneath regions of fast-flowing ice streams (Ritz et al., 2015; Brondex et al., 2017), particularly in the Amundsen Sea Embayment (e.g., Nias et al., 2018; Joughin et al., 2019; Brondex et al., 2019). Despite over 60 years of research on basal sliding (e.g., Weertman, 1957; Lliboutry, 1958a, b, 1959; Budd et al., 1979; Iken, 1981; Iverson et al., 1998; Tulaczyk et al., 2000; Schoof, 2005; Gagliardini et al., 2007; Tsai et al.,

2015; Brondex et al., 2017; Zoet and Iverson, 2020), the sliding law operating on large scales in Antarctica remains a matter of debate.

For ice that slides over the bed, a no-slip boundary condition is inappropriate. Free slip is also unrealistic because basal drag provides significant resistance to sliding wherever the ice is not floating. Instead, a sliding law that relates basal shear stress to sliding speed is needed. Alternative formulations of this sliding law have been proposed, applying to different subglacial

circumstances (e.g., Fig. 1b-f). The frequently used Weertman-type power law (e.g., Weertman, 1957; Arthern et al., 2015; Ritz et al., 2015; Arthern and Williams, 2017; Brondex et al., 2017; Kyrke-Smith et al., 2017; Hank and Tarasov, 2024) considers ice slipping over a rough, hard bed, with ice deforming to pass around large obstacles while bypassing smaller obstacles by pressure melting and regelation (Fig. 1b; Weertman, 1957). In contrast, Lliboutry envisaged discontinuous ice contact with a hard bed, separated by water-filled subglacial cavities (Fig. 1c; Lliboutry, 1958a, b, 1959). Later studies show this cavitation

could lead to an upper bound for basal shear stress, even for fast-sliding glaciers (Iken, 1981; Schoof, 2005), and the upper bound was subsequently included in analytically derived sliding laws (Schoof, 2005; Gagliardini et al., 2007).

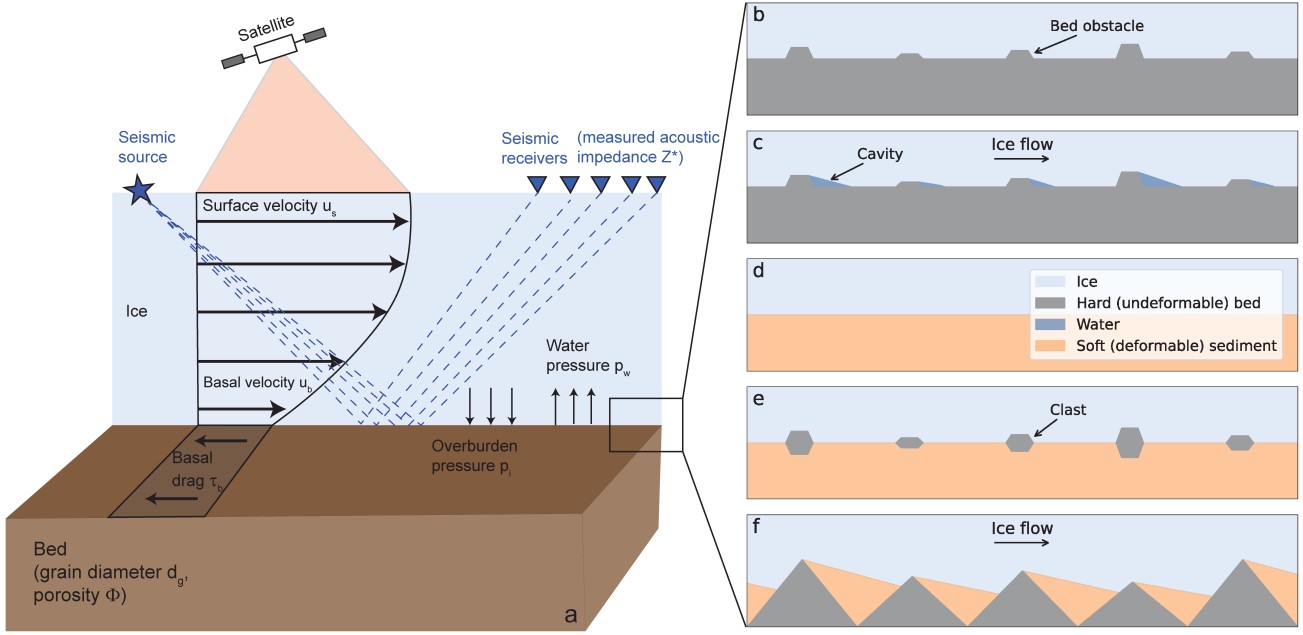

**Figure 1.** Experimental setup (a) and schematics of the bed considered for different sliding laws (b-f). The schematics are not to scale. Details of the experimental setup are outlined in Fig. 2 and the text.

Another type of basal drag law is based on sliding lubricated by a deforming layer of sediment that yields, either in a rate-dependent (viscoplastic) fashion or according to a rate-independent plastic law (Fig. 1d; e.g., Budd et al., 1979; Iverson et al., 1998; Tulaczyk et al., 2000). One such boundary condition is a Coulomb sliding law, for which the basal shear stress

is independent of sliding speed, but varies in proportion to the effective pressure, i.e. the difference between the weight of the overlying ice and the subglacial water pressure. Higher effective pressures lead to greater compression within the granular

sediment. In an alternative formulation, a modification of the Weertman-type power law that accounts for a strong dependence of the basal shear stress on effective pressure found in laboratory experiments has been proposed (Budd et al., 1979).

Ice loss projections, particularly of the Amundsen Sea Embayment, are sensitive to the applied sliding law, with sliding law parameters being a key source of uncertainty (e.g., Gillet-Chaulet et al., 2016; Brondex et al., 2017; Joughin et al., 2019; Brondex et al., 2019; Barnes and Gudmundsson, 2022). Previous approaches constraining the basal properties, i.e. the sliding law parameters, generally rely on remote sensing data and inverse methods (e.g., Arthern et al., 2015; Hoffman et al., 2018; Gudmundsson et al., 2019; Ranganathan et al., 2021) or seismic observations (e.g., Smith et al., 2013; Brisbourne et al., 2017) but lack a direct link between observations and the representation of basal sliding in ice sheet models (Kyrke-Smith et al., 2017).

Here, we present the new BAyesian Sliding Law Inference – Viscous Grain-Shearing (BASLI–VGS) methodology, which enables the quantitative determination of the most appropriate basal sliding law by directly comparing the measured and predicted acoustic impedance, i.e. the product of the compressional wave speed and density of the subglacial material (Fig. 2). The seismic reflection coefficient from the bed is sensitive to the contrast in acoustic impedance between ice and bed. Because the acoustic impedance of ice is known ($3.33 \pm 0.04 \cdot 10^6$ kg m$^{-2}$ s$^{-1}$; Atre and Bentley, 1993), this allows the acoustic impedance of the bed to be recovered from seismic reflection surveys performed in the field (Fig. 1a). The VGS theory of acoustic propagation in granular material (Buckingham, 1997, 2000, 2005, 2007) relates the acoustic impedance to the effective pressure, providing a direct link to the basal sliding law: in most laws, low effective pressure, i.e. high basal water pressure, is associated with fast ice sliding over slippery sediment. As basal water pressure has only been measured directly in a few locations via hot-water drilled boreholes (e.g., Engelhardt et al., 1990; Engelhardt and Kamb, 1997; Lüthi et al., 2002; Smith et al., 2021), it has been difficult to map effective pressure. The new approach provides effective pressure over a much wider area.

## 2 Methods

### 2.1 Linking seismic observations and basal sliding laws

The sliding laws examined in this study (Sec. 2.2) are thought to represent sliding over different subglacial beds (Fig. 1b-f). To infer which of these sliding laws is most probable, we first derive the basal shear stress ($\tau_\mathrm{b}$) and sliding speed ($u_\mathrm{b}$; Fig. 2 and S1) from inverse methods using the Wavelet-based Adaptive-grid Vertically-integrated Ice-sheet-model (WAVI; Arthern et al., 2015; Bradley et al., 2024, Sec. 2.3). The effective pressure ($N$) can then be estimated by rearranging the sliding laws.

The VGS theory (Sec. 2.4) provides a model of acoustic propagation in granular material. Substituting the estimated effective pressure into this model and using independent estimates for grain diameter ($d_\mathrm{g}$) and porosity ($\phi$) from sediment cores (Engelhardt et al., 1990; Stone and Clarke, 1993; Smith et al., 2011; Kirshner et al., 2012; Smith et al., 2014, 2017; Clark et al., 2024, and Smith, unpublished data), provides an estimate of acoustic impedance for each sliding law. The predicted acoustic impedance is then compared to acoustic impedance measurements collected at five sites on Pine Island Glacier (PIG)

in Antarctica (Fig. 4; Brisbourne et al., 2017) by calculating the misfit $\chi^2_{\Theta_i}$ according to

$$70 \quad \chi^2_{\Theta_i} = \frac{1}{N_{\mathrm{d}}} \sum_j^{N_{\mathrm{d}}} \frac{\left(Z_{\Theta_i,j} - Z_j^*\right)^2}{\sigma_j^2}, \qquad (1)$$

where $N_{\mathrm{d}} = 300$ is the number of data points (60 per site, 120 m apart), and $Z_{\Theta_i,j}$ are the acoustic impedance predictions under a given sliding law $i$ and the model parameters $\Theta_i$ (grain diameter and porosity, along with any additional sliding-law-specific parameters; further details in Sec. 2.2 and 2.5). Data points are treated as independent: a sub-sampled data set (every 10th data point) generally yields the same conclusions (Fig. S2 and S3). While there is evidence that PIG is largely underlain by deformable sediments (Muto et al., 2016; Brisbourne et al., 2017), the exact values of $\Theta_i$ are uncertain. Therefore, the misfit $\chi^2_{\Theta_i}$ is systematically assessed across what is considered to be a reasonable parameter space (Sec. 2.5). The model parameters do not vary spatially. $Z_j^*$ and $\sigma_j$ are the acoustic impedance observations and their uncertainties. As an example, all metrics involved in predicting the acoustic impedance and calculating the misfit $\chi^2_{\Theta_i}$ ($u_{\mathrm{b}}$, $\tau_{\mathrm{b}}$, $N$, $Z_{\Theta_i}$, $Z^*$, $\sigma$) are shown for one set of parameter values ($d_{\mathrm{g}} = 0.063$ mm, $\phi = 0.43$, Coulomb friction coefficient $\mu = 0.49$) and the Coulomb sliding law in Fig. S4.

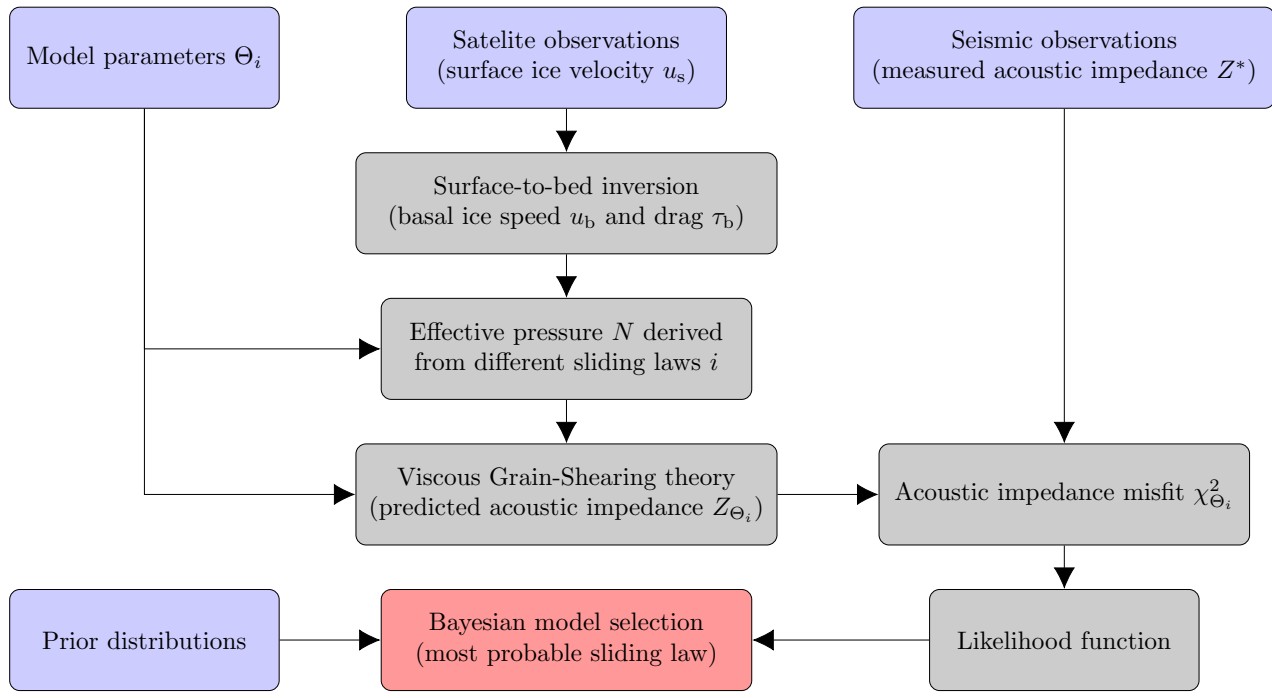

**Figure 2.** Flowchart of the presented BASLI–VGS methodology. Refer to the text for further details.

However, inferring the best-candidate sliding law based solely on the minimum misfit is inadequate, as it does not take into account any prior assessment of the probability of the parameter values used. Instead, we use Bayesian model selection (Sec. 2.5) to identify the most probable sliding law based on all misfits within the parameter space (likelihood function in

Fig. 2). In this framework, the a priori probability of each model, and of particular parameter values within each model is specified by prior distributions. Using Bayes' rule, these prior probabilities are updated using seismic data, to provide posterior probabilities. Ultimately, this allows us to compute the normalized posterior probability of each sliding law, given the seismic observations collected on PIG (Eq. 20). An advantage of the Bayesian approach is that Occam's razor is automatically applied: overly flexible models with a large range or dimension of parameter space are penalized relative to simpler, less flexible models with fewer parameters or tighter bounds upon parameters.

## 2.2 Basal sliding laws

The effective pressure required as input for the VGS theory is determined based on the basal sliding laws described here. Usually, these laws are expressed so that basal drag is a function of sliding speed and effective pressure. To compute effective pressures, these relationships must be inverted, either by explicitly rearranging the equations or by numerical root-finding. For all sliding laws and sites, we ensure the effective pressure does not exceed the ice overburden pressure.

Strictly speaking, the VGS theory used to predict acoustic impedance only applies to granular material (Sec. 2.4). However, while the formation of cavities, for example, is most appropriate for undeformable bed protrusions, larger rock fragments embedded in granular sediment or even fine-grained deformable sediment might play a similar role (Schoof, 2007a, b; Fowler, 2009; Schoof et al., 2012). Therefore, whenever we are using a sliding law initially developed for hard bedrock (Sec. 2.2.3 and 2.2.6), we assume a granular, relatively undeformable material that can not support tangential friction at its interface with the ice (here referred to as *rigid bed*).

### 2.2.1 Fixed effective pressure

The most straightforward approach for estimating the effective pressure $(N)$ – one that does not require the specification of a sliding law – is to assume it is at a fixed fraction of the ice overburden pressure $(p_\mathrm{i})$ everywhere. To contextualize and constrain the results obtained using effective pressures derived from various sliding laws (Sec. 2.2.3 to 2.2.7), we compute the acoustic impedance corresponding to different fractions of the ice overburden pressure, including the two fixed effective pressure endmember scenarios; a lower bound $N = 0\,\mathrm{Pa}$ for which the ice is assumed to be at floatation everywhere, and b) an upper bound, $N = p_\mathrm{i}$, for which the effective pressure is assumed equal to the ice overburden pressure everywhere. These endmembers correspond, respectively, to situations where basal water pressure fully supports the weight of overlying ice or does not support any weight at all.

### 2.2.2 Weertman

The Weertman-type power law (Weertman, 1957) assumes that ice slides perfectly over a rigid bed. A thin water film separating the ice and undeformable bed, allows locally for free slip. The basal drag $\tau_\mathrm{b}$ – resistance to basal motion $u_\mathrm{b}$ – is instead induced by form drag as the ice deforms around the bed obstacles (Fig. 1b). This leads to the relationship

$$\tau_\mathrm{b} = C_\mathrm{W} u_\mathrm{b}^m, \tag{2}$$

where $C_W$ and $m = 1/3$ are, respectively, the Weertman friction parameter and exponent (often related to the creep exponent $n$ in Glen's flow law, $m = 1/n$). As Eq. 2 does not depend on the effective pressure, the Weertman-type power law can not be directly tested within this approach. Instead, we calculate the acoustic impedance for the Budd sliding law.

### 2.2.3 Budd

Laboratory experiments examining temperate ice sliding over bed surfaces with a wide range of roughnesses (e.g., Fig. 1b) indicate that $\tau_b$ exhibits a strong dependence on $N$ (Budd et al., 1979). Consequently, the Weertman-type power law was modified to account for this dependence.

$$\tau_b = C_B u_b^m N^q, \tag{3}$$

where $C_B$ and $q = 1$ are the Budd friction parameter and exponent, respectively.

### 2.2.4 Coulomb

The Coulomb-type plastic rheology sliding law describes ice sliding over soft, deformable sediments (Fig. 1d; Iverson et al., 1998; Tulaczyk et al., 2000).

$$\tau_b = \mu N, \tag{4}$$

where $\mu = \tan(\Phi)$ is the Coulomb friction coefficient and $\Phi$ the till friction angle.

### 2.2.5 Tsai-Budd

A simple sliding law describing basal motion as the combination of ice deformation around and across bed obstacles (Weertman) and deformation of the underlying sediment (Coulomb; Fig. 1e or f; Tsai et al., 2015) takes the form

$$\tau_b = \min[C_W u_b^m, \mu N]. \tag{5}$$

As for the Weertman-type power law itself, Eq. 5 can not be tested in the context discussed here because the Weertman part of the sliding law has no dependence on the effective pressure. To overcome this issue, we replace the Weertman part of Eq. 5 with the Budd sliding law (Eq. 3):

$$\tau_b = \min[C_B u_b^m N^q, \mu N]. \tag{6}$$

### 2.2.6 Schoof

Eq. 2 and 3 neglect Iken's bound induced by water-filled cavities (upper bound of $\tau_b/N$ determined by the maximum up-slope angle of the bed in flow direction $(\beta)$; Fig. 1c; Iken, 1981; Schoof, 2005; Gagliardini et al., 2007). Thus, Schoof (2005) derived a new sliding law incorporating this upper bound. Strictly speaking, the Schoof sliding law only applies to linear ice rheology.

Gagliardini et al. (2007) then numerically extended the relationship to non-linear rheologies. Here we use a generalized form of this sliding law (Brondex et al., 2017):

$$\tau_b = \frac{C_S u_b^m}{(1 + (C_S/(C_{max}N))^{1/m} u_b)^m},$$ (7)

where $C_S$ is the Schoof friction parameter and $C_{max} = \tan\beta$ represents Iken's bound (Iken, 1981; Schoof, 2005).

### 2.2.7 Zoet-Iverson

Based on experiments in which pressurized ice at its melting temperature is slid over a water-saturated till bed, Zoet and Iverson (2020) derived the following sliding law for glaciers on deformable beds (Fig. 1e):

$$\tau_b = N\mu \left(\frac{u_b}{u_b + u_t}\right)^{\frac{1}{p}},$$ (8)

where the transition speed

$$u_t = \frac{\left(\frac{1}{\eta(Ra)^2 k_0^3} + \frac{4C_1}{(Ra)^2 k_0}\right)(N_F N)}{(2 + N_F k)},$$ (9)

$k_0 = \frac{2\pi}{4R}$, and the regelation parameter $C_1 = C_p \frac{K}{L}$. Slightly rearranging Eq. 8 and 9 allows us to numerically determine $N$

$$\tau_b = N\mu \left(\frac{u_b}{u_b + C_{ZI}N}\right)^{\frac{1}{p}},$$ (10)

where

$$C_{ZI} = \frac{\left(\frac{1}{\eta(Ra)^2 k_0^3} + \frac{4C_1}{(Ra)^2 k_0}\right) N_F}{(2 + N_F k)}$$ (11)

is the transition speed coefficient ($u_t$ without the dependence on $N$). All other parameters are listed in Table. 1.

While the mathematical form of the Schoof (Eq. 7) and Zoet-Iverson sliding law (Eq. 10) is very similar, the physical reasoning and interpretation differ. The Schoof sliding law is most applicable for ice sliding over a rigid bed (granular but relatively undeformable material), whereas the Zoet-Iverson sliding law aims to describe ice sliding over a water-saturated till bed (deformable). Similarly, the sliding-law-specific parameters $\mu$ and $C_{max}$ represent distinct physical properties, and, may therefore differ significantly (Sec. 2.5).

### 2.3  Surface-to-bed inversion

Basal shear stress and basal sliding speed are derived using the ice sheet model WAVI, which is vertically integrated but retains an implicit velocity-depth profile (Arthern et al., 2015; Bradley et al., 2024). Data assimilation methods are used to initialise the model into a present-day state (approximately 2015): spatially varying two-dimensional fields of ice stiffness and basal drag are calculated by matching modelled surface velocities with observations of surface velocities (Mouginot et al., 2022),

accumulation rates (Arthern et al., 2006), and thinning rates (Smith et al., 2020). Internal ice temperatures are provided from

| Variable | Description | Value | Unit |
|---|---|---|---|
| $p$ | slip exponent | 5 | - |
| $\eta$ | effective ice viscosity | $3.2 \cdot 10^{12}$ | Pa s |
| $R$ | clast radius | 0.015 | m |
| $a$ | fraction of clast radius that protrudes from bed surface | 0.25 | - |
| $C_p$ | depression of the melting temperature of ice with pressure | $7.4 \cdot 10^{-8}$ | K Pa$^{-1}$ |
| $K$ | mean thermal conductivity of ice and rock | 2.55 | W m$^{-1}$ K$^{-1}$ |
| $L$ | volumetric latent heat of ice | $3 \cdot 10^{8}$ | J m$^{-3}$ |
| $N_f$ | till bearing capacity factor | 33 | - |
| $k$ | till strength reduction resulting from the ice pressure shadow in the lee of clasts | 0.1 | - |

**Table 1.** Parameters used in Eqs. 10 and 11 (supplementary material of Zoet and Iverson (2020) and references therein).

a thermal solve of the BISICLES ice sheet model (Cornford et al., 2013). Full details of the inverse method are detailed in Arthern et al. (2015), and the resulting basal sliding speed and basal shear stress are shown in Fig. S1. In this inversion, the basal drag is identified using the Weertman sliding law. However, the sliding relationship that links basal drag and basal speed can be re-parameterised in terms of any of the selected sliding laws that we test here, as long as neither the basal speed nor the basal drag are altered in this process.

### 2.4 Viscous Grain-Shearing theory

The Viscous Grain-Shearing (VGS) theory (Buckingham, 1997, 2000, 2005, 2007) is used to relate seismic observations to effective pressure (Fig. 2). According to the VGS theory, the elastic deformation under effective pressure that generates frictional resistance also stiffens the sediment and increases the speed of propagation of sound waves. Changes in the speed of sound alter the acoustic impedance ($Z = \rho_s c_p$), the product of the compressional wave speed in the sediment ($c_p$) and density ($\rho_s$). In turn, the acoustic impedance controls the reflection coefficient of seismic energy from the base of the ice sheet. The acoustic propagation model predicts the compressional wave speed ($c_p = \psi[N, d_g, \phi, f_s]$) as a function of effective pressure ($N$), grain diameter ($d_g$), porosity ($\phi$), and seismic frequency ($f_s$). The link between the compressional wave speed and effective pressure predicted by the acoustic model provides an avenue to test whether a given sliding law applies at any location. All other parameters of the acoustic propagation model have been calibrated using acoustic observations of the ocean floor.

The governing equation for the compressional wave speed is

$$c_{\mathrm{p}} = \frac{c_0}{\mathrm{Re}\left[1 + \zeta\left(i\omega T\right)^q g\left(\omega \tau_{\mathrm{p}}\right)\right]^{-1/2}}, \tag{12}$$

where $c_0 = \sqrt{\frac{\kappa_0}{\rho_0}}$ is the sound speed in the absence of grain-to-grain interactions, $\kappa_0 = \left(\frac{\phi}{\kappa_\mathrm{p}} + \frac{1-\phi}{\kappa_\mathrm{g}}\right)^{-1}$ the bulk modulus of the medium, and $\rho_0 = \phi\rho_\mathrm{p} + (1-\phi)\rho_\mathrm{g}$ the bulk density of the medium. The dimensionless grain-shearing coefficient is

$$\zeta = \frac{\gamma_\mathrm{p} + (4/3)\,\gamma_\mathrm{s}}{\rho_0 c_0^2}, \tag{13}$$

where $\gamma_\mathrm{p} = \gamma_\mathrm{p0}\left[\frac{N d_\mathrm{g}}{N_0 d_\mathrm{g0}}\right]^{1/3}$ and $\gamma_\mathrm{s} = \gamma_\mathrm{s0}\left[\frac{N d_\mathrm{g}}{N_0 d_\mathrm{g0}}\right]^{2/3}$ are the compressional and shear rigidity coefficients, respectively. $N_0 = (1-\phi_0)(\rho_\mathrm{g} - \rho_\mathrm{p})gz_0$ is the reference effective pressure. The function

$$g\left(\omega\tau_\mathrm{p}\right) = \left(1 + \frac{1}{i\omega\tau_\mathrm{p}}\right)^{-1+q} \tag{14}$$

accounts for the effect of the viscosity of the molecularly thin layer of pore fluid between contiguous grains ($\nu$). Molecularly thin films become progressively more viscous as they are squeezed, and, therefore, $\nu$ differs significantly from the viscosity of the bulk fluid (Israelachvili, 1986; Luengo et al., 1996; Granick, 1999). The compressional viscoelastic time constant $\tau_\mathrm{p}$ is defined as $\tau_\mathrm{p} = \nu/E$, where $E$ is a spring constant (Buckingham, 2005). The values of $\tau_\mathrm{p}$ used in the VGS theory are visual fits to the *SAX99* experiments (Buckingham, 2007). However, the measurements were taken in 18 to 19 m deep water (Richardson et al., 2001). Therefore, the exerted overburden pressure is $\sim 2$ orders of magnitude smaller (less squeezed) than under PIG (ice thickness of 1500 to 2500 m in tributaries, e.g., Fretwell et al., 2013). While it is apparent that the viscosity of molecularly thin layers increases with the applied pressure (or loading) $p_\mathrm{L}$, the exact relationship between $p_\mathrm{L}$, the thickness of the thin film, and the viscosity $\nu$ is not straightforward (e.g., Israelachvili, 1986; Luengo et al., 1996; Yamada, 2003). Assuming $\nu \propto p_\mathrm{L}$, we set $\tau_\mathrm{p} = 0.012$ s (2 orders of magnitude larger than the value in Buckingham, 2007). However, future studies should further explore the adaptation of the VGS theory from oceanographic to glacial contexts.

$\omega = 2\pi f$ is the angular frequency, $i = \sqrt{-1}$, and Re returns the real part of a complex number. All other parameters are listed in Table 2.

## 2.5 Bayesian model selection

We compare the different sliding laws using Bayes' Rule:

$$P(M_i|D,I) = \frac{P(D,I|M_i)\,P(M_i)}{P(D,I)}, \tag{15}$$

where $D$ represents the data (acoustic impedance observations), $I$ represents the inverted $u_\mathrm{b}$ and $\tau_\mathrm{b}$, and $M_i$ represents the model for sliding law $i$ together with the VGS theory. However, the situation here slightly differs from the routine application of Bayes' rule for inferring model parameters within a single model and is more akin to Bayesian model selection. The main difference for the model selection framework is that the probability space is extended to cover multiple models, each of which has its own parameter space. Since the number of parameters differs between models (e.g., two for the fixed effective pressure scenarios and four for the Zoet-Iverson sliding law) and we aim to compare the posterior probabilities of models $P(M_i|D,I)$, not the joint posterior probability of models and parameters $P(\Theta_i, M_i|D,I)$, we marginalize over the model parameters $\Theta_i$ to

| Variable | Description | Value | Unit |
|---|---|---|---|
| $T$ | arbitrary time introduced to avoid awkward dimensions | 1 | s |
| $q$ | strain hardening index | 0.0851 | - |
| $\kappa_\mathrm{p}$ | bulk modulus of pores | $2.374 \cdot 10^9$ | Pa |
| $\kappa_\mathrm{g}$ | bulk modulus of grains | $3.6 \cdot 10^{10}$ | Pa |
| $\rho_\mathrm{P}$ | density of pore fluid | 1005 | kg m$^{-3}$ |
| $\rho_\mathrm{g}$ | density of grains | 2730 | kg m$^{-3}$ |
| $\gamma_\mathrm{p0}$ | reference compressional coefficient | $3.888 \cdot 10^8$ | Pa |
| $\gamma_\mathrm{s0}$ | reference shear coefficient | $4.588 \cdot 10^7$ | Pa |
| $d_\mathrm{g0}$ | reference grain diameter | $1 \cdot 10^{-3}$ | m |
| $\phi_0$ | reference porosity | 0.377 | - |
| $g$ | acceleration due to gravity | 9.81 | m s$^{-2}$ |
| $z_0$ | reference depth in sediment | 0.3 | m |
| $f_\mathrm{s}$ | seismic frequency | 100 | Hz |
| $\tau_\mathrm{P}$ | compressional viscoelastic time constant | 0.012 | s |

**Table 2.** Parameters used in the VGS theory. The values for $f_\mathrm{s}$ and $\tau_\mathrm{P}$ are based on seismic frequencies in a glaciological context and a scaling analysis of the value used in Buckingham (2007), respectively. All other values are adopted from Buckingham (2005).

retrieve $P(D, I | M_i)$:

$$P(D, I | M_i) = \int_{\Theta_i} P(D, I | \Theta_i, M_i) \, P(\Theta_i | M_i) \, d\Theta_i \tag{16a}$$

$$= \int_{\Theta_i} P(D | I, \Theta_i, M_i) \, P(I | \Theta_i, M_i) \, P(\Theta_i | M_i) \, d\Theta_i. \tag{16b}$$

Assuming the error of the data follows a Gaussian distribution, the likelihood of the acoustic impedance data given the model, its parameters, and the inverted $u_\mathrm{b} - \tau_\mathrm{b}$ is calculated according to

$$P(D | I, \Theta_i, M_i) = \exp\left(-0.5 \chi_{\Theta_i}^2\right). \tag{17}$$

Therefore, the posterior probability of each model $M_i$ is

$$P(M_i | D, I) = \frac{\int_{\Theta_i} \exp\left(-0.5 \chi_{\Theta_i}^2\right) P(I | \Theta_i, M_i) \, P(\Theta_i | M_i) \, d\Theta_i \, P(M_i)}{\sum_{j=1}^{n} \int_{\Theta_j} \exp\left(-0.5 \chi_{\Theta_j}^2\right) P(I | \Theta_j, M_j) \, P(\Theta_j | M_j) \, d\Theta_j \, P(M_j)}. \tag{18}$$

The prior information from the inverted $u_\mathrm{b} - \tau_\mathrm{b}$ (not used to constrain $P(\Theta_i | M_i)$) can be directly incorporated into an updated prior using Bayes' rule:

$$P(\Theta_i | I, M_i) = \frac{P(I | \Theta_i, M_i) \, P(\Theta_i | M_i)}{P(I | M_i)}, \tag{19}$$

where $P(I|M_i) = \int_{\Theta_i} P(I|\Theta_i, M_i) \, P(\Theta_i|M_i) \, d\Theta_i$ is a normalization term. Eq. 18 can then be written as

$$P(M_i|D, I) = \frac{\int_{\Theta_i} \exp\left(-0.5\chi^2_{\Theta_i}\right) P(\Theta_i|I, M_i) \, d\Theta_i \, P(M_i|I)}{\sum_{j=1}^{n} \int_{\Theta_j} \exp\left(-0.5\chi^2_{\Theta_j}\right) P(\Theta_j|I, M_j) \, d\Theta_j \, P(M_j|I)}, \tag{20}$$

where we use a prior $P(M_i|I) = 1/n$ that considers each sliding law equally probable, with $n$ being the number of sliding laws considered. Posterior probabilities calculated using $P(M_i) = 1/n$, i.e. without the normalization through $P(I|M_i)$ in Eq. 19, are shown in Fig. S6.

Finally, the prior distributions for all model parameters $P(\Theta_i|M_i)$ need to be defined. The prior distributions for all individual parameters are shown in Fig. 3. The combination of multiple individual priors creates a model's parameter space $\Theta_i$ and determines the model prior $P(\Theta_i|M_i)$. Since the parameter space differs between the models (number of individual parameters (dimensions) as well as number of tested parameter values), we ensure $\int_{\Theta_i} P(\Theta_i|M_i) \, d\Theta_i = 1$ for all models. This normalization reflects the fact that once a model has been chosen, the parameters of that model must lie somewhere within its parameter space with certainty. This is self-evident and automatically applies Occam's Razor, penalizing models with a larger parameter space compared to less flexible models. The key idea of Occam's Razor is that a balance between goodness of fit and model flexibility is desirable, but we emphasise that no special manipulations are required to enforce this balance in the Bayesian approach.

When constructing the parameter space $\Theta_i$, the prior distributions of individual parameters are treated as independent of one another. Although physical relationships among some of these parameters have been described in the literature, the formulation of a coupled prior remains challenging, as these relationships are often convoluted by other properties. For instance, the porosity is generally inversely related to the mean (or median) grain size, but this relationship is convoluted by, e.g., the particle size uniformity (e.g., Wang et al., 2017; Atapour and Mortazavi, 2018; Gupta and Ramanathan, 2018; Díaz-Curiel et al., 2024). As the Bayesian model selection framework already downweights extreme parameter combinations (e.g., high porosity and large grain size) through the chosen independent prior distributions, and because the minimum misfit and most probable parameters are generally consistent with, e.g., the porosity-grain size relationship described in the literature (e.g., Díaz-Curiel et al., 2024), we do not expect a significant change in the posterior probabilities.

Various literature estimates inform the examined parameter ranges and corresponding prior distributions. The porosity prior (Fig. 3a) is derived from borehole data and seismic experiments from Ice Stream B and C, West Antarctica (Blankenship et al., 1987; Engelhardt et al., 1990; Atre and Bentley, 1993), borehole data from Trapridge Glacier, Yukon Territory, Canada (Stone and Clarke, 1993), marine sediment cores from the Amundsen Sea Embayment (Table S1; Smith et al., 2011, 2014, 2017), sediment recovered from beneath Rutford Ice Stream, West Antarctica (Table S1; Smith, unpublished data), as well as the porosity of sands and glass beads used to validate the VGS (Buckingham, 2014; Lee et al., 2016, and references therein). As the latter do not directly relate to a glacial context, we assign these higher porosities a lower probability. The porosity estimates from seismic experiments (Blankenship et al., 1987; Atre and Bentley, 1993) assume no significant dependence on effective pressure and are employed as an independent comparison rather than to directly inform the prior.

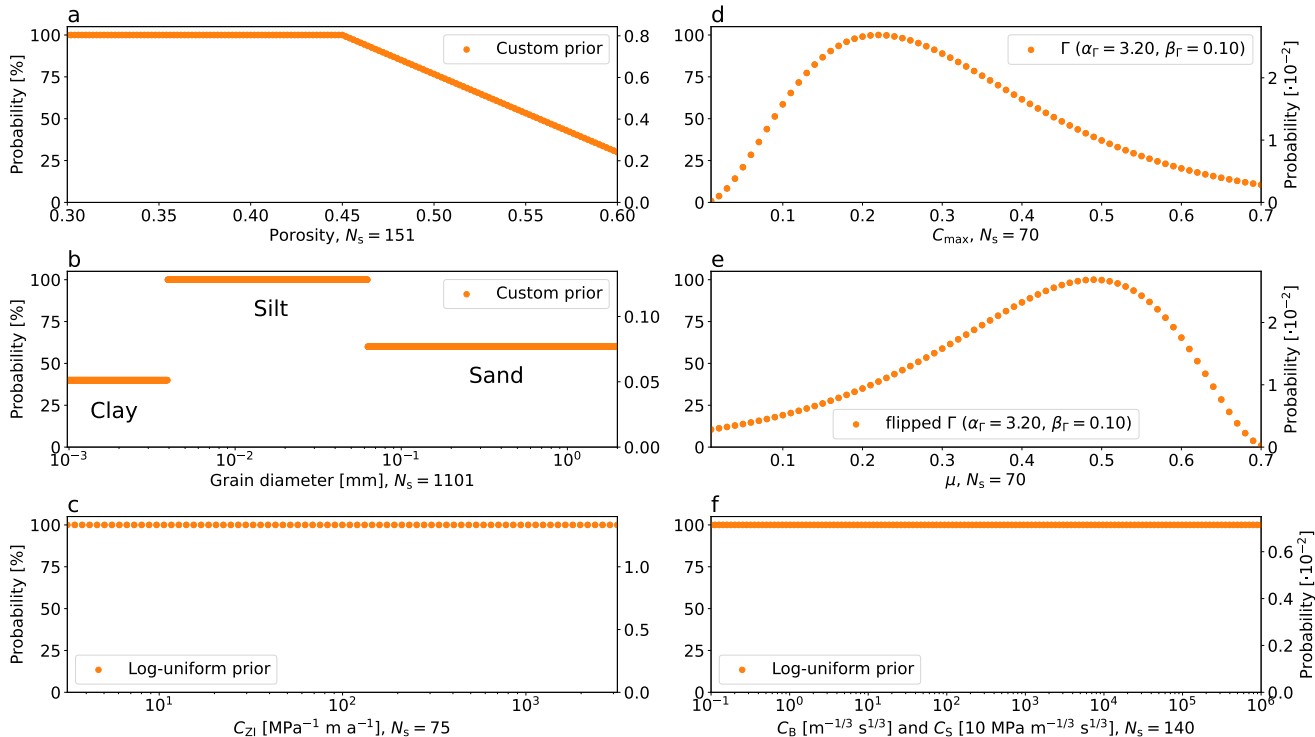

**Figure 3.** Prior distributions for all model parameters $\Theta$. $N_s$ is the sampling size. The left and right axes show the scaled probability (0 to 100 %) and actual probability used (depends on $N_s$), respectively.

The grain diameter prior (Fig. 3b) is based on sediment cores collected in the Amundsen Sea Embayment, particularly Pine Island Bay (Table S1; Kirshner et al., 2012; Smith et al., 2011, 2014, 2017; Clark et al., 2024) and the Rutford ice stream (Table S1; Smith, unpublished data). We differentiate between Clay ($< 1/256$ mm), Silt ($\geq 1/256$ mm and $\leq 1/16$ mm), and Sand ($> 1/16$ mm). The prior is then derived from the relative fractions of these grain-size classes.

The transition speed coefficient ($C_{\mathrm{ZI}}$) values reported in the initial publication of the Zoet-Iverson sliding law range from 56.36 to 363.52 MPa$^{-1}$ m yr$^{-1}$ (Zoet and Iverson, 2020). A later study using the same bed material (Horicon till sourced from the same location) but with plowing clasts removed uses the same parameters (given in Table S1 of Zoet and Iverson, 2020) except for a smaller clast radius $R = 0.0045$ m (instead of $R = [0.015, 0.030]$ m), leading to $C_{\mathrm{ZI}} = 1120.17$ MPa$^{-1}$ m yr$^{-1}$ (Fig. S4 in Hansen et al., 2024). Given these significant uncertainties and that $C_{\mathrm{ZI}}$ depends on several other uncertain parame-

ters, a log-uniform prior covering the range 3.16 to 3155.76 MPa$^{-1}$ m yr$^{-1}$ was chosen (Fig. 3c).

The $C_{\mathrm{max}}$ prior (Fig. 3d) is based on bed topography beneath PIG retrieved from Bedmap2 data (Fig. S7 and S8; Fretwell et al., 2013) as well as autonomous underwater vehicle (AUV) data collected downstream of Thwaites Glacier (Graham et al., 2022) and under the Thwaites Eastern Ice Shelf (Wåhlin, unpublished data; Fig. S9 and S10). While shear resistance is most likely built at scales smaller than the resolution of Bedmap2, the bed roughness and therefore the actual relevant scale are

less clear and likely vary spatially. As these smaller scales are not explicitly represented by the basal drag derived from our inversion, it is not straightforward to determine the $C_{\mathrm{max}}$ prior directly from the small-scale AUV data. Therefore, we align the highest probability in the $C_{\mathrm{max}}$ prior with the steepest Bedmap2 bed angles and incorporate even steeper bed angles at smaller scales through a more gradual decline towards higher $C_{\mathrm{max}}$ values (Sec. S6.2).

$\mu$ is a frequently used parameter and its prior (Fig. 3e) aims to capture the overall distribution within the glaciological community (e.g., Savage et al., 2000; Tulaczyk et al., 2000; Cuffey and Paterson., 2010; Iverson, 2010; Tsai et al., 2015; Brondex et al., 2017). Note that although $C_{\mathrm{max}}$ and $\mu$ serve similar roles in, e.g., the Schoof and Zoet-Iverson sliding law, they represent distinct physical properties and are thus assigned separate prior distributions (Sec. 2.2.7).

As $C_{\mathrm{B}}$ and $C_{\mathrm{S}}$ are positive scaling coefficients that may vary over several orders of magnitude, even within the same glacial catchment (Budd et al., 1984; Larour et al., 2012; Favier et al., 2014; Arthern et al., 2015; Brondex et al., 2017; Gladstone et al., 2017), a log-uniform prior was chosen for these parameters (Fig. 3f).

Due to the computational cost of the grid search, we currently limit the model parameter space $\Theta_i$ to 4D. For example, we do not consider variations in the exponents $m$, $q$, and $p$ (Sec. 2.2). However, computationally more efficient methods, such as Monte Carlo algorithms, can be explored in future studies to simultaneously vary more than four parameters.

## 3   Results and Discussion

### 3.1   Minimum acoustic impedance misfit comparable for all sliding laws examined

Based on a previous study examining the same acoustic impedance data (Kyrke-Smith et al., 2017) and due to the smoothing effect of the inversion (1 km horizontal grid resolution), we do not expect to capture acoustic impedance variations for each individual data point but rather the general trend across the five data sites. Given this context, all sliding laws reasonably match the acoustic impedance observations when using the parameter values yielding the minimum misfit across all data sites (Fig. 4). However, for some sliding laws, the minimum misfit parameter values are at the limits of the likely range (e.g., extremely small grain diameter ($d_{\mathrm{g}} = 0.003$ mm) for the Budd sliding law). While the minimum misfit might correspond to a rather unlikely parameter value, a narrow band of similarly small misfits spans a more reasonable parameter range, indicating some indistinctness in the selected minimum misfit parameter values. This is a key characteristic of the misfit distribution in all of our experiments. As an example, Fig. 5 shows how the misfit varies with the three model parameters $d_{\mathrm{g}}$, $\phi$, and $\mu$ when using a Coulomb sliding law. The same plots for all other sliding laws with a maximum 3D parameter space are shown in Fig. S11 to S22.

### 3.2   Ice dynamics of Pine Island Glacier governed by Coulomb-type sliding

To consider the misfit distribution across the entire parameter range and any prior assessment of the probability of the parameter values used, we infer the best-candidate sliding law based on Bayesian model selection. The Coulomb sliding law has the highest posterior probability of all sliding laws tested (increase of 27.5 % relative to the prior; Fig. 6). However, the Schoof

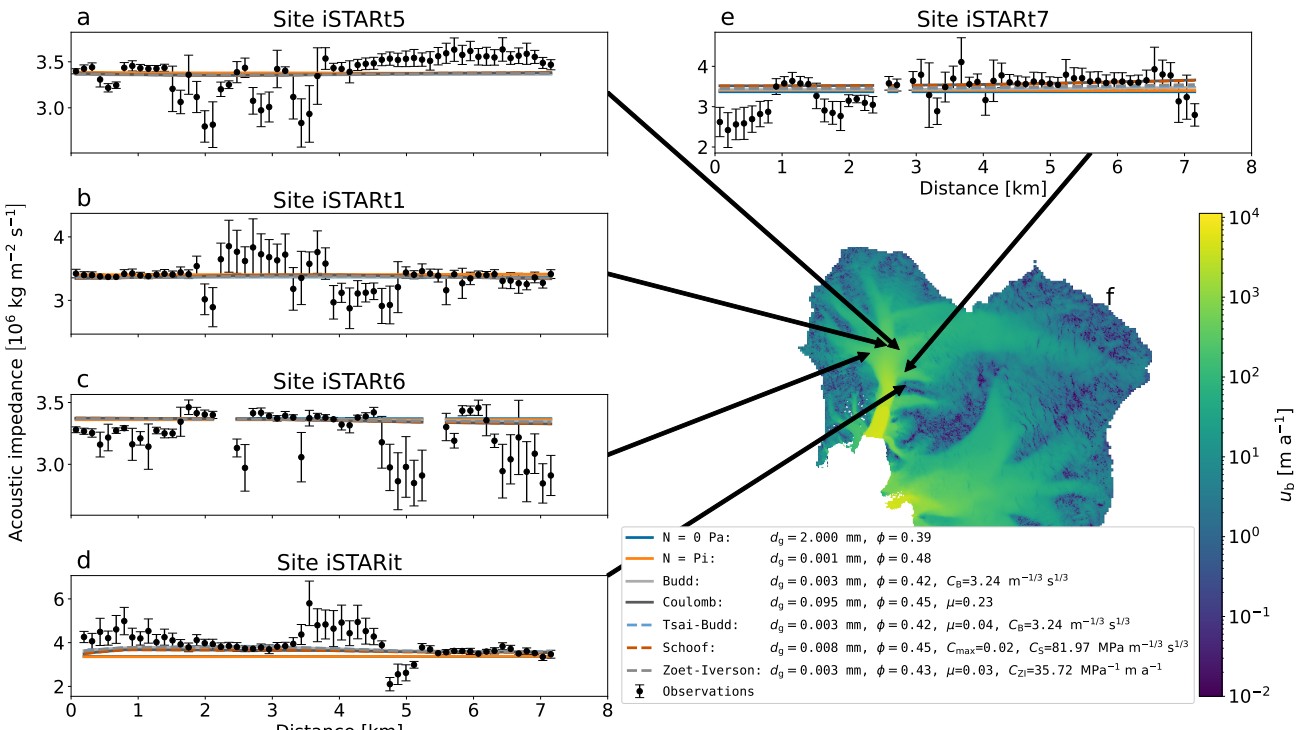

**Figure 4.** (a-e) Acoustic impedance observations (Brisbourne et al., 2017) compared with the model predictions based on different sliding laws when using the minimum misfit model parameters shown in the legend. The observational uncertainties are shown as error bars. The model parameters are grain diameter ($d_g$), porosity ($\phi$), Budd friction parameter ($C_B$), Coulomb friction coefficient ($\mu$), Iken's bound ($C_{\max}$), and transition speed coefficient ($C_{ZI}$; see Sec. 2.2 for details). (f) Basal sliding speed in the Amundsen Sea Embayment (from inversion; Sec. 2.3). The arrows mark the location of the data sites. Except for site iSTARit, all data were collected on fast-flowing tributaries of PIG (Brisbourne et al., 2017).

and Zoet-Iverson sliding laws show a similarly strong increase, hindering the determination of a single-best sliding law. The Tsai-Budd sliding law exhibits the smallest increase (4.8 %) out of all the laws incorporating a Coulomb friction term of the form $\mu N$ or $C_{\max} N$. Nonetheless, the increase in posterior probability for all sliding laws incorporating a Coulomb friction term suggests this is a desirable property of a sliding law. In comparison, the Budd sliding law, without the $\mu N$ modification of the Tsai-Budd law, performs worse (0.8 % decrease). The fixed effective pressure endmember scenario that assumes $N = p_i$ everywhere performs worst of all, leading to the smallest posterior probability (83.4 % decrease). The endmember scenario with $N = 0$ Pa everywhere yields the highest posterior probability of all fixed effective pressure experiments (4.1 % increase; see also Fig. S23).

The relatively high posterior probabilities of sliding laws incorporating a Coulomb friction term and the $N = 0$ Pa endmember scenario are consistent with the widespread occurrence of deformable sediment under the fast-flowing tributaries of PIG (Brisbourne et al., 2017). Furthermore, the high probabilities of these sliding laws align with previous studies identifying

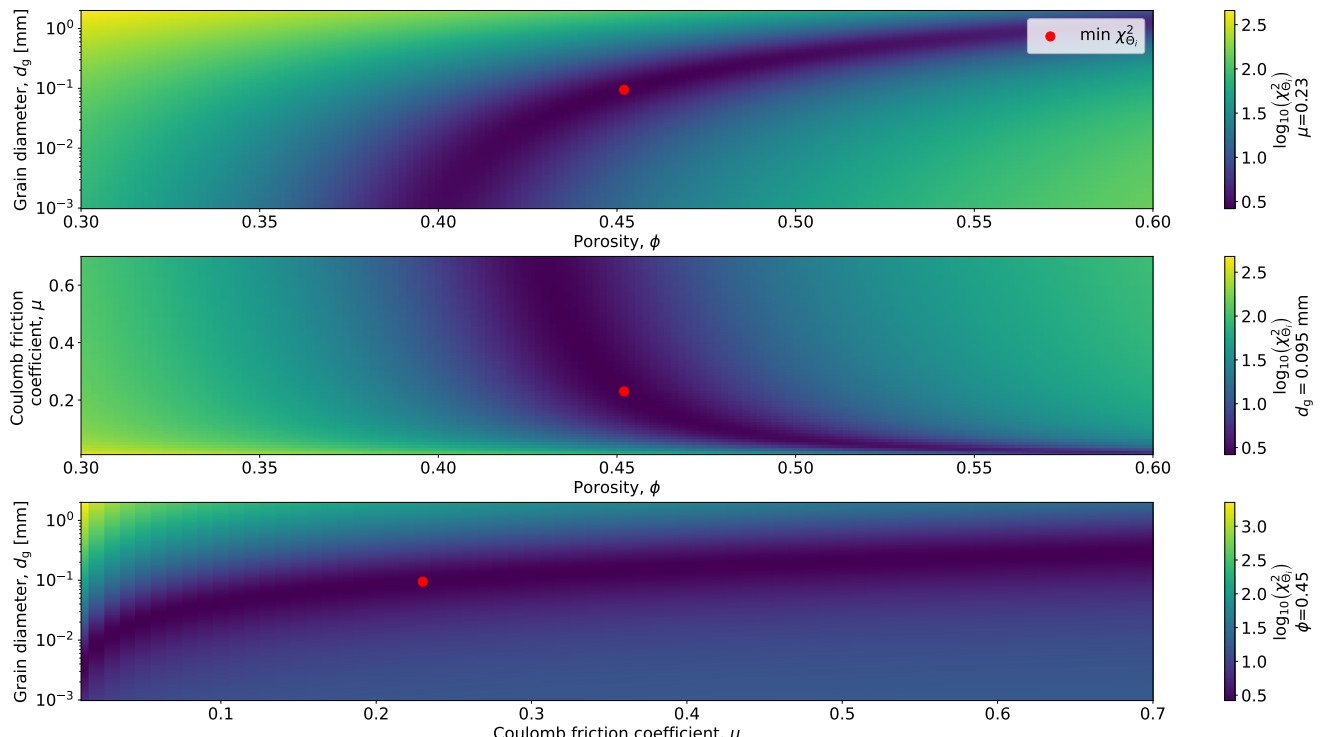

**Figure 5.** Variations of the misfit $\chi^2_{\Theta_i}$ with the three model parameters grain diameter ($d_g$), porosity ($\phi$), and Coulomb friction coefficient ($\mu$) under a Coulomb sliding law. For the parameter not shown, the value yielding the minimum misfit is used and denoted next to the colorbar of the corresponding panel. The red dots mark the minimum misfit.

(quasi-)plastic deformation of the underlying sediment as the primary mode of sliding for PIG (Gillet-Chaulet et al., 2016; Joughin et al., 2019). While the sensitivity of grounding-line retreat patterns and mass loss projections to the choice of sliding law is high (Brondex et al., 2019), determining the exact implications of using a (quasi-)plastic sliding law on glacier behaviour through prognostic simulations for all sliding laws and parameter values is out of the scope of this study. In general, sliding laws representing a (quasi-)plastic rheology lead to higher sea level rise contributions (Ritz et al., 2015; Gillet-Chaulet et al., 2016; Brondex et al., 2019).

### 3.3 Effect of prior distributions on most probable model parameters

As for the minimum misfit model parameters, the predicted acoustic impedance under the model parameters with the highest posterior probability generally agrees with the observations within uncertainties for all sliding laws tested (Fig. S24). In the remainder of this paper, we refer to the model parameters with the highest posterior probability as the maximum a posteriori (MAP) parameters. When examining the MAP parameters in more detail (Fig. S24), the effect of the chosen prior distributions is evident. Although covering the full range within this size classification, the MAP grain diameter for all sliding laws is Silt-

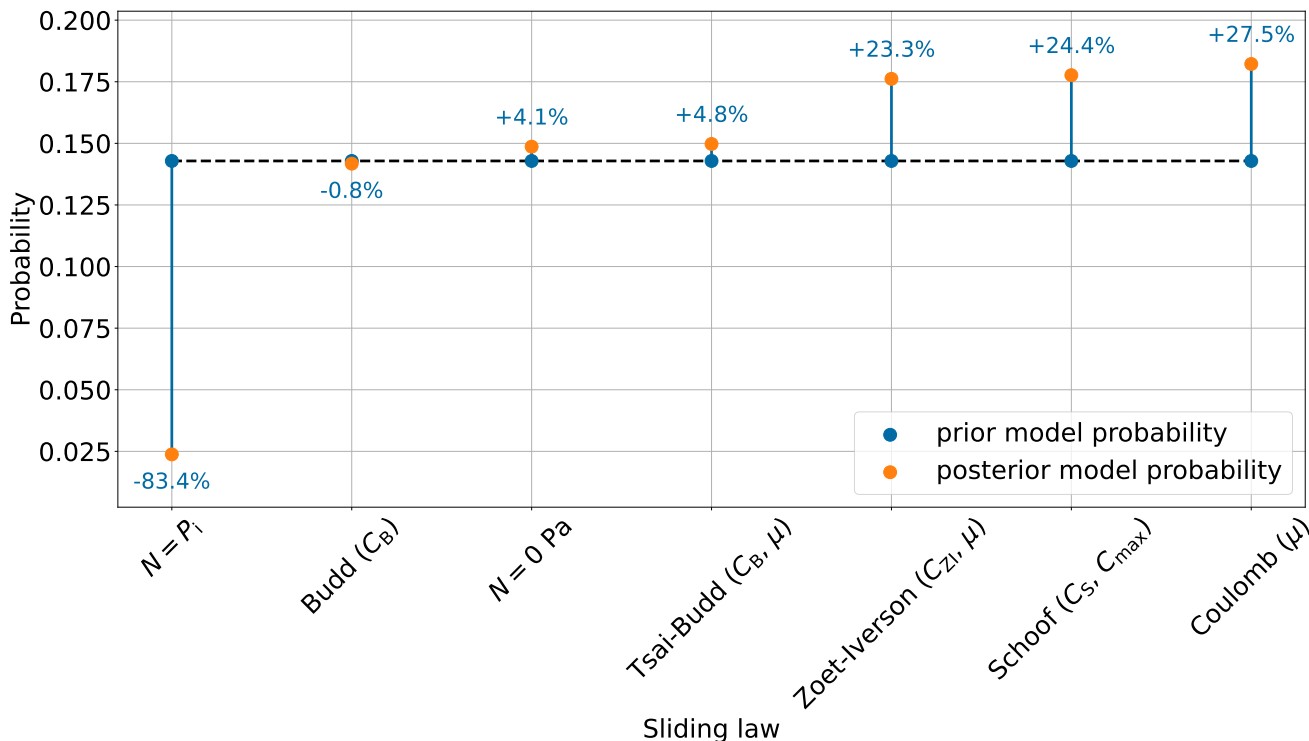

**Figure 6.** Normalized probabilities (Eq. 20) of all sliding laws examined in this study given the acoustic impedance observations collected on PIG. The prior model probability is $P(M_i|I) = 1/n$, with $n$ being the number of models examined (blue circles; dashed horizontal line visualizes equal prior probability). To obtain the posterior model probability, we marginalized over all corresponding model parameters $\Theta_i$, encompassing the acoustic propagation model parameters ($d_g$ and $\phi$) and any additional sliding-law-specific parameters (denoted in brackets). No sliding law parameter was varied for the two fixed effective pressure endmember scenarios $N = p_i$ and $N = 0$ Pa. The prior distributions for all parameters are shown in Fig. 3. The blue vertical lines and numbers indicate the change in probability.

sized (highest prior probability; Fig. 3). The MAP porosities ($0.39$ to $0.44$) are at the upper end of the high-prior probability

range ($\phi = [0.3, 0.45]$) for all sliding laws except the fixed effective pressure endmember scenario $N = p_i$ ($\phi = 0.55$; Fig. S24), indicating comparatively porous sediments beneath PIG. Similarly, the MAP values of the unique sliding law parameters without a log-uniform prior distribution ($\mu$ and $C_{max}$) are in the vicinity of the highest prior probability.

Even if we use log-uniform prior distributions for scaling coefficients and uniform priors for other parameters, the sliding laws incorporating a Coulomb friction term still yield the highest probabilities (Fig. S25). This demonstrates the robustness of

our key result against variations in prior distributions.

### 3.4 Low effective pressure across most of Amundsen Sea Embayment

Excluding the fixed effective pressure scenarios, the predicted effective pressure for the MAP model parameters is generally below $0.1$ MPa ($1$ bar) for the $4$ sites within fast-flowing tributaries (Fig. S26). The relatively high probability of the $N = 0$ Pa

endmember scenario (Fig. 6 and S23) further supports a low effective pressure. This is in agreement with previous effective
pressure estimates derived from, e.g., shear wave velocities (Blankenship et al., 1987), borehole water level measurements
(Engelhardt et al., 1990; Engelhardt and Kamb, 1997; Lüthi et al., 2002; Smith et al., 2021), and the widespread presence of
active subglacial lakes (Gray et al., 2005; Fricker et al., 2007; Smith et al., 2009).

Site iSTARit, located between two tributaries, has higher effective pressures (0.1 to 1 MPa), with the effective pressure
derived from the Coulomb sliding law being $\sim 0.1$ MPa. We hypothesize that the higher effective pressure and resulting
increased basal drag at this site hinder basal sliding.

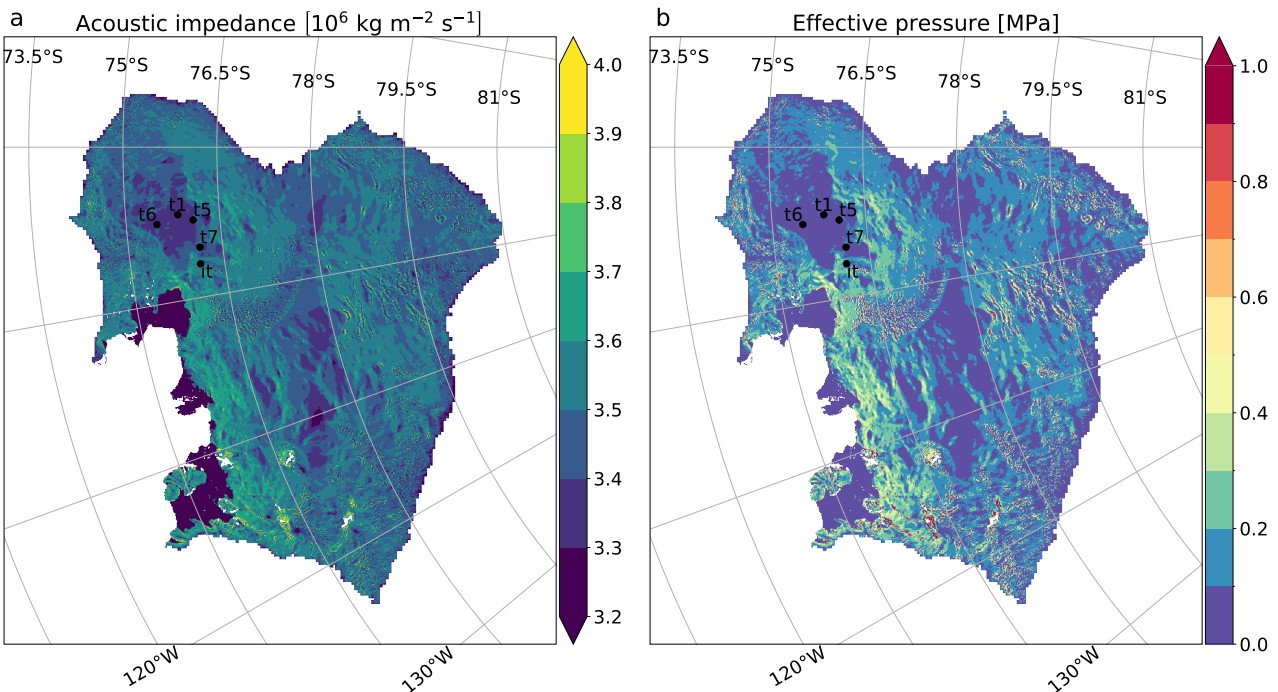

**Figure 7.** Predicted acoustic impedance (a) and effective pressure (b) in the Amundsen Sea Embayment when using a Coulomb sliding
law with the MAP (highest posterior probability) model parameters ($d_\mathrm{g} = 0.063$ mm, $\phi = 0.43$, and $\mu = 0.49$). The black dots mark the
locations of the seismic observation sites.

Retrieving the effective pressure for the Coulomb sliding law with the MAP parameters across the whole Amundsen Sea
Embayment indicates the effective pressure is generally below $0.5$ MPa (Fig. 7b). Being closely related to the basal drag,
this map represents the slipperiness of the bed, with areas of low effective pressure being susceptible to fast retreat. However,
the effective pressure map is based on a spatially uniform $\mu$ obtained from five sites in PIG and does not capture (local)
dynamic subglacial systems as, e.g., represented by a subglacial hydrology model. Furthermore, using only the Coulomb sliding
law with the MAP parameters neglects the probabilities of other sliding laws and parameter values. Therefore, the provided
effective pressure map should be used with caution. Following the Bayesian framework to determine the most probable effective

pressure map by weighting the individual maps for all sliding laws and parameter values, incorporating spatially variable model parameters, as well as applying BASLI–VGS in regions characterized by higher basal heterogeneity (e.g., Thwaites Glacier),
should be explored in future studies.

## 4   Conclusions

In this study, we present the new BASLI–VGS approach that directly relates measured and predicted acoustic impedance data. Since the predicted acoustic impedance depends on the effective pressure, an ice sheet sliding law and its parameters can be inferred, subsequently enabling the derivation of an effective pressure map. While the current conclusions are primarily
based on seismic data over soft sediments, the presented methodology can be readily applied to any acoustic impedance data collected in glacial environments underlain by granular material. For the seismic data collected on fast-flowing tributaries of Pine Island Glacier, the acoustic propagation model predicts the observed acoustic impedance within uncertainties. Inferred effective pressures are generally below $0.5\,\mathrm{MPa}$ across most of the Amundsen Sea Embayment and below $0.1\,\mathrm{MPa}$ within fast-flowing tributaries of Pine Island Glacier. Bayesian model selection identifies Coulomb behaviour as the most probable mode
of sliding, potentially increasing sea level rise contributions from the Amundsen Sea Embayment. To minimize uncertainties in sea level rise projections, the sliding law used in large-scale ice sheet models should, therefore, approach Coulomb behaviour in fast-flowing regions.

*Code availability.*   The main code has been attached to the submission of this manuscript. After acceptance for publication, the code will be made available in a public archive.

*Data availability.*   Data is available upon request from the corresponding author. After acceptance for publication, the data will be made available in a public archive.

*Code and data availability.*   TEXT

*Sample availability.*   TEXT

*Video supplement.*   TEXT

*Author contributions.* KH, RJA, and CRW conceptualized the ideas behind this study. AMB, JAS, and AW prepared, respectively, the seismic data, sediment core data, and AUV data for use in this study. KH prepared the experimental design, ran the model, and analyzed the results with input from all authors. All authors contributed to the interpretation of the results and writing of the paper.

*Competing interests.* The authors have no competing interests.

*Disclaimer.* TEXT

*Acknowledgements.* The authors thank the members of the International Thwaites Glacier Collaboration (ITGC), particularly the ITGC Geophysical Habitat of Subglacial Thwaites (GHOST) team for fruitful discussion. We also thank Ronan S. Agnew and Kelly A. Hogan for their support in the interpretation of acoustic impedance and autonomous underwater vehicle (AUV) data, respectively. Finally, we thank two anonymous reviewers and the handling topic editor, Adam Booth, for their constructive comments.

This work was funded by the GHOST project, a component of the International Thwaites Glacier Collaboration (ITGC). Support from National Science Foundation (NSF: Grant PLR 1738934) and Natural Environment Research Council (NERC: Grant NE/S006672/1), with logistics provided by NSF-U.S. Antarctic Program and NERC-British Antarctic Survey. ITGC Contribution No. ITGC-142. Additional support was provided by the ITGC MELT project (NSF Grant 1739003 and NERC Grant NE/S006656/1) and Natural Environment Research Council (NERC: Grant NE/G014159/1 and NE/R016038/1). The Ran AUV was financed by Knut and Alice Wallenberg Foundation.

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
