# Peer review of "Inferring the ice sheet sliding law from seismic observations: A Pine Island Glacier case study"

_EGUsphere, 2025_

## Referee Comment (RC1)

**Review for EGUSPHERE-2025-764**

The central idea of this paper is clever and thoughtfully executed. The authors use satellite-derived surface velocities to invert for basal shear stress and sliding speed, applying a suite of commonly used basal sliding laws that either implicitly or explicitly depend on effective pressure. While the inversion of satellite data using various slip laws is not itself novel, having been employed in numerous prior studies, this paper introduces a direct, testable link between effective pressure predictions and independent constraints derived from seismic acoustic impedance data. By leveraging Buckingham's Viscous Grain-Shearing (VGS) theory, informed by marine sediment analogs and supported by site-specific porosity and grain size priors, the authors generate effective pressure predictions from seismic observations that can be compared directly to the inversions. While the acoustic model carries inherent uncertainties, the authors mitigate these through a Bayesian model selection framework, which allows for robust comparison of competing sliding laws while accounting for parameter uncertainty.

I find this to be a well-written and conceptually significant paper that advances our ability to constrain basal conditions of Antarctic ice streams. The methodology sets a precedent for integrating seismic and glaciological observations into a coherent, probabilistic framework—providing new potential for mapping the spatial (and eventually temporal) distribution of basal effective pressure. Importantly, the results add further weight to the relevance of Coulomb-style or regularized-Coulomb sliding laws, which continue to be debated within the modeling community.

My primary critiques do not detract from the paper's novelty or value, but they do point to natural directions for future works. I do believe they should be briefly addressed in the text.

- The VGS theory, while well-motivated, is adapted from oceanographic contexts and relies on assumptions about pressure dependence that have not been directly tested under glacial conditions.
- The use of independent prior distributions may oversimplify the relationships among subglacial sediment properties, particularly where physical coupling through compaction or consolidation is expected.
- The study draws on seismic data from only five sites, which limits the spatial resolution and generalizability of the inferred effective pressure fields. It is not surprising, at least to me, that PIG exhibits Coulomb-like behavior as I'm not aware of any studies that contradict this. As a point of curiosity, I am interested to see how this methodology performs in other environments where basal conditions are debated, such as the interior of the Greenland Ice Sheet or alpine glaciers (obviously outside the scope of this study!)

I believe the authors work transparently with the available data and generally acknowledge the limitations. I view this paper as an important step forward in linking geophysical observations to ice sheet model parameterizations and recommend it for publication following minor revisions that clarify the scope of applicability and emphasize caveats in interpreting the effective pressure maps. *I have a few minor comments which I detail below:*

**Comments related to the prior distributions:**

I would like to see more detail on the logic behind the formulation of the custom prior distributions. While I appreciate the justification for the $C_{max}$ prior shown in the Supplement, the distributions for porosity and grain size are less clear. Since these appear to be new compilations from the literature, it would be helpful to include the underlying data (in the Supplement would be sufficient) and to show how those priors were constructed from the compiled observations. Additionally, in cases where porosity was estimated from active seismic data (e.g., Blankenship et al., 1987), it's worth noting that those estimates assumed no dependence on effective stress. This could introduce some circularity when those values are used to constrain priors in a model that explicitly incorporates effective stress. Clarifying these points would strengthen the study.

Secondly, grain size, porosity, and effective stress are not independent in natural systems, but are physically coupled through compaction, consolidation, and sediment mechanics. If I understand the methodology correctly, parameter sets were sampled independently from their prior distributions, grain size and porosity, for example, and then used to calculate effective stress via Buckingham's VGS theory. However, relationships between these variables have been described in the sediment mechanics literature and impose constraints on what combinations are physically reasonable. I am concerned that treating them as statistically independent in the prior sampling may lead to internally inconsistent sediment states. While the Bayesian framework helps downweight poor-fitting combinations, would explicitly incorporating physically based constraints or coupled priors could improve the robustness of the analysis in a meaningful way?

Regarding $u_t$, I respect the uncertainty that leads the authors to use a log-uniform prior, but as I recall, the Zoet-Iverson slip law includes a prediction for $u_t$ based on sediment properties (most notably grain size) which already has a relatively narrow range in this study. Given that, it doesn't seem reasonable to expect $u_t$ values near $10^4$ m/yr as equally likely as, say $10^2$? There are also at least two other studies I can recall that provide calculated values of $u_t$ in different configurations: Helanow et al. (2020; DOI: 10.1126/sciadv.abe7798) for sliding over rough, rigid beds and Hansen et al. (2024; DOI/10.1029/2023GL107681) for frozen sediments over till. Some discussion of this would be helpful, as it's not clear whether the wide prior range used here is physically justified.

It would be helpful to emphasize more clearly in the introduction or discussion that the method presented here is primarily applicable to soft-bedded glacier systems, since the acoustic impedance contrast relies on wave propagation through a granular medium. This is an important distinction, especially considering that some of the tested sliding laws were originally formulated for rigid or mixed bed topographies. I think an open question remains in glaciology regarding how these different sliding laws apply across regions with spatially heterogeneous basal conditions (e.g., Maier et al., 2021, https://doi.org/10.5194/tc-15-1435-2021). The result that a fast-flowing, soft-bedded glacier like Pine Island Glacier exhibits Coulomb-style sliding is not surprising to me, given the preponderance of experimental and field evidence in the literature. But in light of continued and recent discussion in the literature (e.g., Law et al., 2024, https://doi.org/10.48550/arXiv.2407.13577) it would be worth emphasizing the both the utility and the limitation of the geophysical datasets to constrain the slip law.

Excellent work overall!

---

## Author Comment (AC1)

**Author's response to Anonymous Referee 1 Comment 1**

July 14, 2025

**General comments**

We thank the referee for their constructive comments. A point-by-point reply is reported below, with referee comments in orange and our replies in black.

[The VGS theory, while well-motivated, is adapted from oceanographic contexts and relies on assumptions about pressure dependence that have not been directly tested under glacial conditions.]
and
[The study draws on seismic data from only five sites, which limits the spatial resolution and generalizability of the inferred effective pressure fields. It is not surprising, at least to me, that PIG exhibits Coulomb-like behavior as I'm not aware of any studies that contradict this. As a point of curiosity, I am interested to see how this methodology performs in other environments where basal conditions are debated, such as the interior of the Greenland Ice Sheet or alpine glaciers (obviously outside the scope of this study!)]
While we agree that the VGS theory needs further testing under glacial conditions, we did adjust the compressional viscoelastic time constant $\tau_{\mathrm{p}}$ to account for the difference in exerted overburden pressure. We are currently working on applying the same methodology to Thwaites Glacier, the results of which will be presented in a follow-up publication. A brief discussion of the points above will be added to the revised manuscript.

**Specific comments**

[I would like to see more detail on the logic behind the formulation of the custom prior distributions. While I appreciate the justification for the Cmax prior shown in the Supplement, the distributions for porosity and grain size are less clear. Since these appear to be new compilations from the literature, it would be helpful to include the underlying data (in the Supplement would be sufficient) and to show how those priors were constructed from the compiled observations. Additionally, in cases where porosity was estimated from active seismic data (e.g., Blankenship et al., 1987), it's worth noting that those estimates assumed no dependence on effective stress. This could introduce some circularity when those values are used to constrain priors in a model that explicitly incorporates effective stress. Clarifying these points would strengthen the study.]
A supplementary table outlining the grain size and porosity data will be added to the revised manuscript. The porosity prior was primarily informed by borehole data and sediment cores. Similar to the porosity of sands and glass beads used to validate the VGS, the seismic estimates from, e.g., Blankenship et al. [1987] function in a supplementary capacity. We will clarify this in the revised manuscript and incorporate the referee's suggestion.

[The use of independent prior distributions may oversimplify the relationships among subglacial sediment properties, particularly where physical coupling through compaction or consolidation is expected.]
and

[Secondly, grain size, porosity, and effective stress are not independent in natural systems, but are physically coupled through compaction, consolidation, and sediment mechanics. If I understand the methodology correctly, parameter sets were sampled independently from their prior distributions, grain size and porosity, for example, and then used to calculate effective stress via Buckingham's VGS theory. However, relationships between these variables have been described in the sediment mechanics literature and impose constraints on what combinations are physically reasonable. I am concerned that treating them as statistically independent in the prior sampling may lead to internally inconsistent sediment states. While the Bayesian framework helps downweight poor-fitting combinations, would explicitly incorporating physically based constraints or coupled priors could improve the robustness of the analysis in a meaningful way?]
In general, the porosity is inversely related to the mean (or median) grain size, but this relationship is convoluted by other properties such as the particle size uniformity [e.g., Wang et al., 2017, Atapour and Mortazavi, 2018, Gupta and Ramanathan, 2018, Díaz-Curiel et al., 2024]. While it is correct that the parameter sets were sampled independently, and using coupled priors would improve the robustness of the analysis for our most extreme parameter combinations (e.g., high porosity and large grain size), the relationship between porosity and grain size outside these extreme parameter combinations, and therefore the formulation of such a coupled prior, is less clear. As the Bayesian framework already downweights the extreme parameter combinations through the chosen independent prior distributions (as correctly identified by the referee), and the minimum misfit and MAP parameters are generally consistent with the porosity-grain size relationship described in the literature [e.g., Díaz-Curiel et al., 2024], we do not expect a significant change in the posterior probabilities. We will add a brief discussion of this to the revised manuscript.

[Regarding ut, I respect the uncertainty that leads the authors to use a log-uniform prior, but as I recall, the Zoet-Iverson slip law includes a prediction for ut based on sediment properties (most notably grain size) which already has a relatively narrow range in this study. Given that, it doesn't seem reasonable to expect ut values near $10^4$ m/yr as equally likely as, say $10^2$? There are also at least two other studies I can recall that provide calculated values of ut in different configurations: Helanow et al. (2020; DOI: 10.1126/sci-adv.abe7798) for sliding over rough, rigid beds and Hansen et al. (2024; DOI/10.1029/2023GL107681) for frozen sediments over till. Some discussion of this would be helpful, as it's not clear whether the wide prior range used here is physically justified.]
Zoet and Iverson [2020] report $u_{t,noN}$ values in the range 56.36 to 363.52 MPa$^{-1}$ m yr$^{-1}$. Because Hansen et al. [2024] use the same bed material (Horicon till sourced from same location) but with plowing clasts removed, they use the model parameters given in Table S1 in Zoet and Iverson [2020] except for a smaller clast radius $R = 0.0045$ m (instead of $R = 0.015$ m or $R = 0.030$ m), leading to $u_{t,noN} = 1120.17$ MPa$^{-1}$ m yr$^{-1}$. Given these significant uncertainties and that $u_{t,noN}$ depends on several other uncertain parameters, we argue that $u_{t,noN}$ is best represented by a log-uniform prior (currently covering the range 3.16 to 3155.76 MPa$^{-1}$ m yr$^{-1}$). We will include these additional details in the revised manuscript. Note that the regularised Coulomb law used in Helanow et al. [2021] is not the same as in Zoet and Iverson [2020].

[It would be helpful to emphasize more clearly in the introduction or discussion that the method presented here is primarily applicable to soft-bedded glacier systems, since the acoustic impedance contrast relies on wave propagation through a granular medium. This is an important distinction, especially considering that some of the tested sliding laws were originally formulated for rigid or mixed bed topographies. I think an open question remains in glaciology regarding how these different sliding laws apply across regions with spatially heterogeneous basal conditions (e.g., Maier et al., 2021, https://doi.org/10.5194/tc-15-1435-2021). The result that a fast-flowing, soft-bedded glacier like Pine Island Glacier exhibits Coulomb-style sliding is not surprising to me, given the preponderance of experimental and field evidence in the literature. But in light of continued and recent discussion in the literature (e.g., Law et al., 2024, https://doi.org/10.48550/arXiv.2407.13577) it would be worth emphasizing the both the utility and the limitation of the geophysical datasets to constrain the slip law.]
The referee is correct that, strictly speaking, the Viscous Grain-Shearing theory only applies to granular material. However, as outlined in detail in our response to the second referee, whenever we are using

a sliding law originally formulated for hard beds (e.g., Budd, Schoof), we assume a granular, relatively undeformable material that cannot support tangential friction at its interface with the ice (here referred to as *rigid bed*). We will clarify this in the revised manuscript.

We agree with the referee that spatially heterogeneous basal conditions remain an open research question and spatially variable parameters (grain size, porosity, as well as sliding law parameters) should thus be explored in future studies. We will comment on this in the revised manuscript.

**References**

Hadi Atapour and Ali Mortazavi. The effect of grain size and cement content on index properties of weakly solidified artificial sandstones. *Journal of Geophysics and Engineering*, 15(2):613, feb 2018. doi: 10.1088/1742-2140/aaa14a. URL https://dx.doi.org/10.1088/1742-2140/aaa14a.

D. D. Blankenship, C. R. Bentley, S. T. Rooney, and R. B. Alley. Till beneath ice stream b. 1. properties derived from seismic travel times. *Journal of Geophysical Research*, 92:8903–8911, 1987. ISSN 01480227. doi: 10.1029/JB092iB09p08903.

Jesús Díaz-Curiel, Bárbara Biosca, Lucía Arévalo-Lomas, David Paredes-Palacios, and María J. Miguel. On the Influence of Grain Size Compared with Other Internal Factors Affecting the Permeability of Granular Porous Media: Redefining the Permeability Units. *Lithosphere*, 2024(1): lithosphere_2023_231, 01 2024. ISSN 1941-8264. doi: 10.2113/2024/lithosphere_2023_231. URL https://doi.org/10.2113/2024/lithosphere_2023_231.

Aniket Gupta and A. L. Ramanathan. Grain texture as a proxy to understand porosity, permeability and density in Chandra Basin, India. *SN Applied Sciences*, 1(1):1, 2018. ISSN 2523-3971. doi: 10.1007/s42452-018-0001-3. URL https://doi.org/10.1007/s42452-018-0001-3.

D. D. Hansen, K. L. P. Warburton, L. K. Zoet, C. R. Meyer, A. W. Rempel, and A. G. Stubblefield. Presence of frozen fringe impacts soft-bedded slip relationship. *Geophysical Research Letters*, 51(12):e2023GL107681, 2024. doi: https://doi.org/10.1029/2023GL107681. URL https://agupubs.onlinelibrary.wiley.com/doi/abs/10.1029/2023GL107681. e2023GL107681 2023GL107681.

Christian Helanow, Neal R. Iverson, Jacob B. Woodard, and Lucas K. Zoet. A slip law for hard-bedded glaciers derived from observed bed topography. *Science Advances*, 7:2–9, 2021. ISSN 23752548. doi: 10.1126/sciadv.abe7798.

Ji-Peng Wang, Bertrand François, and Pierre Lambert. Equations for hydraulic conductivity estimation from particle size distribution: A dimensional analysis. *Water Resources Research*, 53(9):8127–8134, 2017. doi: https://doi.org/10.1002/2017WR020888. URL https://agupubs.onlinelibrary.wiley.com/doi/abs/10.1002/2017WR020888.

Lucas K. Zoet and Neal R. Iverson. A slip law for glaciers on deformable beds. *Science*, 368, 4 2020. ISSN 10959203. doi: 10.1126/science.aaz1183.

---

## Author Comment (AC2)

**Author's response to Anonymous Referee 2 Comment 1**

July 14, 2025

**General comments**

We thank the referee for their constructive comments. A point-by-point reply is reported below, with referee comments in orange and our replies in black. We agree with the specific referee comments not listed here and will revise the manuscript accordingly. Specific comments that merely repeat points already addressed in the referee's general comments are also not listed here.

[Bayesian approaches are generally used to determine the posterior probability density function (PDF) of model parameters, given prior information and constraining observations. For each sliding law, you thus obtain a posterior PDF as a function of the three chosen varying parameters. To obtain a probability for a sliding law, you integrate the posterior PDF over the three-dimensional parameter space (if I understand correctly). This last step is not justified at all in the manuscript while it is critical as all conclusion are based on this. It is not clear to me that a higher integrated probability over the whole parameter space makes a sliding law more likely than another. For example, a model with high but localized maximum PDF can have a lower score than smaller maximum PDF spread on a larger domain of the parameter space. The way the sliding law probability is calculated clearly needs theoretical background. This is critical for the paper as the data does not bring significant difference in misfit and thus data-based likelihood.]

The referee is correct to point out that Bayesian approaches are generally used to determine the posterior probability density function (PDF) of model parameters, given prior information and constraining observations. The situation that we consider here is slightly different, however, and is more akin to Bayesian model selection than the routine application of Bayes' rule for a single model. The main difference for the model selection framework is that the probability space is extended to cover multiple models, each of which has its own parameter space. Apart from that distinction, standard manipulations of probability are used, including Bayes' rule, marginalisation, and normalisation [e.g., Jaynes, 2003].

It is true that we do not justify these standard manipulations, but we will emphasise in the revised manuscript that nothing unusual is happening beyond a straightforward extension of the probability space to acknowledge the possibility of multiple different models.

The statement $\int_{\Theta_i} P(\Theta_i|M_i)\, d\Theta_i = 1$ says that once a model has been chosen, the parameters of that model must lie somewhere in its parameter space with certainty. This is self-evident. By performing this normalisation for each model, we take advantage of the well-known capacity of Bayesian model selection to automatically apply Occam's Razor. Overly flexible models with a large range or dimension of parameter space are penalised relative to simpler, less flexible models with fewer parameters or tighter bounds upon parameters.

The referee questions the marginalisation over the model parameters $\Theta_i$ (as expressed by equation 16), but this is standard because we wish to compare the posterior probabilies of models $P(M_i|D)$, not the joint posterior probability of models and parameters $P(\Theta_i, M_i|D)$. Note that $P(\Theta_i|M_i)$ is a conditional probability like any other and can be manipulated using standard rules of probability (e.g., Equation 16).

Unlike the more standard application of Bayes' rule, each model in the Bayesian model selection framework has its own particular parameter space, and this parameter space can be of any dimension. The two fixed effective pressure endmember scenarios have a 2D parameter space (grain size and porosity).

All other sliding laws have a 3D or 4D parameter space (one or two additional sliding law parameters). The number of values examined for these additional sliding law parameters varies across different sliding laws. Having an additional dimension or a larger number of values examined effectively increases the chance of obtaining a good fit to the data, and this is compensated for appropriately in the Bayesian approach. The key idea is that a balance between goodness of fit and model flexibility is desirable, but we emphasise that no special manipulations are required to enforce this balance in the Bayesian approach, as it emerges quite naturally. We will add these further details about the theoretical background of Occam's razor to the revised manuscript.

[I do not understand why you are limiting your parameter space to three varying parameters. I suspect this is because you do an exhaustive grid search to build the posterior PDF. There are simple methods such as the Monte Carlo algorithm, that can be used to efficiently calculate the posterior PDF in cases where the parameter space is large. This would be easy to implement in your case where the forward model is fast to compute. This limitation forces you to calculate two different probabilities for some sliding laws (Schoof and Zoet-Iverson), where you arbitrarily fix one of the sliding parameters. This makes no sense to me, especially when you assume $\mu = C_{\max} = 0.5$ without justification (when varying ut or $C_s$). The PDF should be built with varying all relevant parameters together.]

The referee is correct that we limited the parameter space to three dimensions due to the computational cost of the grid search. Therefore, our previous results more precisely identified which of the models with three or fewer dimensions best represent the measured acoustic impedance data. We have since expanded the parameter space to four dimensions for all sliding laws with previously two different three-dimensional representations (Tsai-Budd, Schoof, and Zoet-Iverson) and will discuss these results in the revised manuscript (see Fig. 1 for a subset of the examined sliding laws; not all 4D results are available at the time of upload of this response). We agree that methods to simultaneously explore even more parameters, e.g. different exponents, should be explored in future studies, and we will comment on this limiting factor in the revised manuscript.

[Figure]

Figure 1: Normalized probabilities of a subset of the examined sliding laws, including the Tsai-Budd sliding law when simultaneously varying $C_B$ and $\mu$. Note that not all 4D results are available at the time of upload of this response.

[I do not agree with the claim you are testing the Weertman law. You are simply testing the hypothesis of uniform effective pressure which as nothing to do with the Weertman law. If you want to say that the Weertman law is not appropriate you should show that the inverted $\tau_b$ as a function of $u_b$ does not match a power law. A figure showing the inverted $\tau_b$ as a function of $u_b$ is missing in the manuscript in any case.]
It is true that we can not directly test the Weertman sliding law or any sliding law, for that matter, that has no effective pressure dependence. To clarify this, we will refrain from using the term *Weertman-type endmember scenarios* and instead refer to these experiments as *fixed effective pressure endmember scenarios*. However, the effective pressure is only uniform in the $N = 0$ Pa case, as all other fractions of the ice overburden pressure vary spatially due to the dependence on ice thickness.

The relation between the inverted $\tau_b$ and $u_b$ is the same for all sliding laws and, therefore, provides by itself no information on which sliding law is most appropriate. Therefore, we refrain from adding a figure showing the inverted $\tau_b$ as a function of $u_b$.

[I do not think you are able to distinguish which of the stress bounded sliding law perform better when the result is so dependent of the design of the Bayesian approach. You also hide that the Schoof law is almost the exact same law as the Zoet-Iverson law. You can indeed write the equation (7) of the manuscript in this form:]

$$\tau_b = C_{\max} N \left( \frac{u_b}{u_b + \left( \frac{C_{\max}}{C_s} N \right)^{\frac{1}{m}}} \right)^m \tag{1}$$

This is very similar to Zoet and Iverson with $p = 1/m$, $\mu = C_{\max}$ and $u_t = (C_{\max}/C_s N)^{1/m}$. The only difference is that $u_t$ is a function of $N^{1/m}$ in the Schoof formulation and a function of $N$ in Zoet-Iverson. We agree that it is difficult to select a single-best sliding law due to the small differences in posterior probabilities between some of the sliding laws incorporating a Coulomb friction term (Coulomb, Tsai-Budd, Schoof, Zoet-Iverson). For this reason, we focus on the distinction between the Coulomb-type and non-Coulomb-type sliding laws (fixed $N$ endmember scenarios and Budd). We will state this more clearly in the revised manuscript.

Generally speaking, whenever we are using a sliding law originally formulated for hard beds (e.g., Budd, Schoof), we assume a granular, relatively undeformable material that cannot support tangential friction at its interface with the ice (here referred to as *rigid bed*). The formation of cavities, for example, is most appropriate for undeformable bed protrusions, but larger rock fragments embedded in granular sediment or even fine-grained deformable sediment might play a similar role [Schoof, 2007a,b, Fowler, 2009, Schoof, 2012]. The basal drag for rigid beds is dominated by the deformation of ice around bed obstacles (form drag). In contrast, basal conditions dominated by skin drag are covered by the soft bed (deformable sediment) sliding laws (e.g., Coulomb, Zoet-Iverson).

While the form of the Schoof and Zoet-Iverson sliding law is indeed very similar, the physical reasoning and interpretation differ. As described above, the Schoof sliding law is most applicable for ice sliding over a rigid bed (granular but relatively undeformable material). It allows for the formation of cavities and incorporates Iken's bound [$C_{\max} = \tan\beta$; Iken, 1981, Schoof, 2005]. $\beta$ is the maximum up-slope angle of the bed in flow direction. In contrast, the Zoet-Iverson sliding law aims to describe ice sliding over a water-saturated till bed (deformable). $\mu = \tan(\Phi)$ is the Coulomb friction coefficient and $\Phi$ the till friction angle. Thus, the two sliding laws represent different basal conditions, and $\mu$ and $C_{\max}$ describe different physical properties. We will add a brief discussion of the similar mathematical form but different physical reasoning and interpretation to the revised manuscript.

[Posterior PDF are not shown, it would be usefull to have them in some figures to discuss the influence of prior PDF.]
We examine the influence of the prior distribution by applying (log-)uniform priors to all parameters (Fig. 6 vs. S22). As showing 2D planes of the 3D or 4D posterior PDF might be misleading, and to keep the manuscript concise (7 additional plots would be required), we refrain from adding additional map plots.

[Given the resolution of Bedmap-2, the estimation of $C_{\max}$ based on basal topography observation does not make any sense. Even if the inversion is performed at the kilometer scale, the relevant scale at which to estimate $C_{\max}$ is the meter scale, as this is the scale at which shear resistance is built. Also, the impedance model is based on the assumption of a sediment layer, which is inconsistent with the estimation of $C_{\max}$ based on the hard-bed theory. I do not see why $\mu$ and $C_{\max}$ should have different priors, given that they play the same role in the friction law. Doing so favours one sliding law based on unjustified choices.]
While we agree that shear resistance is most likely built at scales smaller than the resolution of Bedmap-2, the bed roughness and therefore the actual relevant scale are less clear and likely vary spatially. However, these smaller scales will not be explicitly represented by the basal drag derived from the inversion. Therefore, the $C_{\max}$ prior is determined by a combination of the Bedmap-2 bed angles and autonomous underwater vehicle data (2 m resolution), taking smaller resolutions into consideration. To clarify this, we will provide further details on the determination of the $C_{\max}$ prior in the main manuscript (previously primarily described in supplement section S4).

The referee is correct that, strictly speaking, the Viscous Grain-Shearing theory only applies to granular material. However, as outlined in detail above, this is consistent with our definition of rigid beds (granular but relatively undeformable material). Furthermore, glacier beds, e.g. the bed beneath Thwaites glacier, often do not support the clear differentiation between rigid beds and soft sediments assumed in the derivation of sliding laws. Instead, the bed might consist of a thin, deformable sediment layer draped over a rigid bed or alternating patches of sediment and rigid bed. Ultimately, the goal of this study is to find the basal sliding parameterisation that best captures the basal conditions identified by the acoustic impedance measurements. As the referee pointed out, by assuming $C_{\max} = \mu$, we would effectively be testing two very similar sliding laws, which undermines this objective. Following this logic, and since $\mu$ and $C_{\max}$ describe different physical properties, there is no reason why the two parameters should have the same prior.

[The title is a too strong statement compared to what you are actually able to infer. Furthermore you focus only on Pine Island glacier, not all Antarctica. I would propose instead: "Evidence of stress bounded friction law at Pine Island Glacier (Antarctica) inferred from seismic observations."]
We agree that the title is misleading and we will adjust it in the revised manuscript. However, while we only infer the sliding law for Pine Island Glacier, the methodology developed here can be applied to basically any acoustic impedance measurement collected on the Antarctic Ice Sheet.

**Specific comments**

[L123 - By doing this you are not testing the Tsai law anymore .... This is the Budd part which make Tsai less likely in your result. I would remove the Tsai law as you cannot really test it.]
It is correct that we examine the Tsai-Budd instead of the Tsai sliding law itself and we will clarify this in the revised manuscript. However, due to its unique concept and mathematical form, the Tsai-Budd sliding law provides valuable insights, and we, therefore, prefer to keep it as part of the analysis.

[L125 - why this value ?]
and
[L137 - you should mention that you fixed Cmax=0.5 when varying Cs and explain why]
and
[Fig. 5 - why 0.5 ?? The best $\mu$ is 0.23. I expect you would use the best value found when varying $u_t$]
$\mu = 0.5$ is the Coulomb friction coefficient with the highest prior probability. Following this logic, we initially set $C_{\max} = 0.2$ (the value with the highest prior probability). However, this value led to a high percentage of incompatible $u_\mathrm{b}-\tau_\mathrm{b}$ pairs and we, therefore, increased it to $C_{\max} = 0.5$.

Using $\mu = 0.23$ would favour the Zoet-Iverson sliding law compared to the other sliding laws, as this information only becomes available through running the experiments. For the Tsai-Budd sliding law when varying $\mu$, we previously relied on the referee's suggested approach as our prior knowledge about $C_\mathrm{B}$ is limited (log-uniform prior). However, all of the above is no longer an issue, since we now vary all four

parameters simultaneously.

[L150 - you could call this parameter differently as it is not a speed anymore...something like "transition speed coefficient" and write it $1/C_{zi}$. this would be more consistent with the Schoof law where a similar coefficient is equal to $(C_{max}/C_s)$. So you would have in the schoof law: $u_t = (C_{max}/C_s * N)^{1/m}$ and in the Zoet-Iverson law: $u_t = (1/C_{zi})N$.]

We agree that "transition speed coefficient" is a better description of $u_{t,noN}$ and will revise the manuscript as follows: $u_{t,noN} = C_{ZI} = u_t/N$.

[L217 - Based on what the prior values are chosen?]

Following the suggestion of the first referee, a table containing detailed information on the porosity and grain size data as well as further details for the priors of the specific sliding law parameters (e.g., $C_{max}$) will be added to the revised manuscript.

[Fig. 6 - why Schoof($C_s$) is not here ?]

Schoof($C_s$) was not included here because of the large number of incompatible $u_b$–$\tau_b$ pairs. We will add further information regarding this issue to the revised manuscript.

[L266 - you should give the MAP parameters]

The MAP parameters are listed in Fig. S21 and S23. We will add a reference to these figures here.

[L299 - this comes from the friction law, not directly the modeled impedance. It should be clear.]

The effective pressure is calculated using the friction law, but the friction law parameter used in this calculation is inferred from the acoustic impedance misfit. We will clarify this in the revised manuscript.

**References**

A. C. Fowler. Instability modelling of drumlin formation incorporating lee-side cavity growth. *Proceedings of the Royal Society A: Mathematical, Physical and Engineering Sciences*, 465(2109):2681–2702, 2009. doi: 10.1098/rspa.2008.0490. URL https://royalsocietypublishing.org/doi/abs/10.1098/rspa.2008.0490.

Almut Iken. The effect of the subglacial water pressure on the sliding velocity of a glacier in an idealized numerical model. *Journal of Glaciology*, 27:407–421, 1981. ISSN 0022-1430. doi: 10.3189/s0022143000011448.

E. T. Jaynes. *Probability Theory: The Logic of Science*. Cambridge University Press, 2003.

Christian Schoof. The effect of cavitation on glacier sliding. *Proceedings of the Royal Society A: Mathematical, Physical and Engineering Sciences*, 461:609–627, 2005. ISSN 14712946. doi: 10.1098/rspa.2004.1350.

Christian Schoof. Cavitation on Deformable Glacier Beds. *SIAM Journal on Applied Mathematics*, 67(6):1633–1653, 2007a. doi: 10.1137/050646470. URL https://doi.org/10.1137/050646470.

Christian Schoof. Pressure-dependent viscosity and interfacial instability in coupled ice–sediment flow. *Journal of Fluid Mechanics*, 570:227–252, 2007b. doi: 10.1017/S0022112006002874.

Christian Schoof. Marine ice sheet stability. *Journal of Fluid Mechanics*, 698:62–72, 2012. ISSN 00221120. doi: 10.1017/jfm.2012.43.

---

## Author Response (AR1)

**Author's response to Referee Comments**

September 2, 2025

**1 Author's response to Anonymous Referee 1 Comment 1**

**General comments**

We thank the referee for their constructive comments. A point-by-point reply is reported below, with referee comments in orange, our replies in black, and the revisions in light blue. References to figures, tables, and sections in our replies (black) refer to the original manuscript, whereas those in the revisions (light blue) correspond to the revised version.

[The VGS theory, while well-motivated, is adapted from oceanographic contexts and relies on assumptions about pressure dependence that have not been directly tested under glacial conditions.]
and
[The study draws on seismic data from only five sites, which limits the spatial resolution and generalizability of the inferred effective pressure fields. It is not surprising, at least to me, that PIG exhibits Coulomb-like behavior as I'm not aware of any studies that contradict this. As a point of curiosity, I am interested to see how this methodology performs in other environments where basal conditions are debated, such as the interior of the Greenland Ice Sheet or alpine glaciers (obviously outside the scope of this study!)]
While we agree that the VGS theory needs further testing under glacial conditions, we did adjust the compressional viscoelastic time constant $\tau_\mathrm{p}$ to account for the difference in exerted overburden pressure. We are currently working on applying the same methodology to Thwaites Glacier, the results of which will be presented in a follow-up publication. We added However, future studies should further explore the adaptation of the VGS theory from oceanographic to glacial contexts. and [...] applying BASLI–VGS in regions characterized by higher basal heterogeneity (e.g., Thwaites Glacier), should be explored in future studies.

**Specific comments**

[I would like to see more detail on the logic behind the formulation of the custom prior distributions. While I appreciate the justification for the Cmax prior shown in the Supplement, the distributions for porosity and grain size are less clear. Since these appear to be new compilations from the literature, it would be helpful to include the underlying data (in the Supplement would be sufficient) and to show how those priors were constructed from the compiled observations. Additionally, in cases where porosity was estimated from active seismic data (e.g., Blankenship et al., 1987), it's worth noting that those estimates assumed no dependence on effective stress. This could introduce some circularity when those values are used to constrain priors in a model that explicitly incorporates effective stress. Clarifying these points would strengthen the study.]
A supplementary table outlining the grain size and porosity data was added to the revised manuscript (Table S1). Furthermore, we added The porosity estimates from seismic experiments (Blankenship et al., 1987; Atre and Bentley, 1993) assume no significant dependence on effective pressure and are employed as an independent comparison rather than to directly inform the prior.

[The use of independent prior distributions may oversimplify the relationships among subglacial sediment properties, particularly where physical coupling through compaction or consolidation is expected.]

and

[Secondly, grain size, porosity, and effective stress are not independent in natural systems, but are physically coupled through compaction, consolidation, and sediment mechanics. If I understand the methodology correctly, parameter sets were sampled independently from their prior distributions, grain size and porosity, for example, and then used to calculate effective stress via Buckingham's VGS theory. However, relationships between these variables have been described in the sediment mechanics literature and impose constraints on what combinations are physically reasonable. I am concerned that treating them as statistically independent in the prior sampling may lead to internally inconsistent sediment states. While the Bayesian framework helps downweight poor-fitting combinations, would explicitly incorporating physically based constraints or coupled priors could improve the robustness of the analysis in a meaningful way?]

In general, the porosity is inversely related to the mean (or median) grain size, but this relationship is convoluted by other properties such as the particle size uniformity (e.g., Wang et al., 2017; Atapour and Mortazavi, 2018; Gupta and Ramanathan, 2018; Díaz-Curiel et al., 2024). While it is correct that the parameter sets were sampled independently, and using coupled priors would improve the robustness of the analysis for our most extreme parameter combinations (e.g., high porosity and large grain size), the relationship between porosity and grain size outside these extreme parameter combinations, and therefore the formulation of such a coupled prior, is less clear. As the Bayesian framework already downweights the extreme parameter combinations through the chosen independent prior distributions (as correctly identified by the referee), and the minimum misfit and MAP parameters are generally consistent with the porosity-grain size relationship described in the literature (e.g., Díaz-Curiel et al., 2024), we do not expect a significant change in the posterior probabilities. We added this discussion to the revised manuscript: When constructing the parameter space $\Theta_i$, the prior distributions of individual parameters are treated as independent of one another. Although physical relationships among some of these parameters have been described in the literature, the formulation of a coupled prior remains challenging, as these relationships are often convoluted by other properties. For instance, the porosity is generally inversely related to the mean (or median) grain size, but this relationship is convoluted by, e.g., the particle size uniformity (e.g., Wang et al., 2017; Atapour and Mortazavi, 2018; Gupta and Ramanathan, 2018; Díaz-Curiel et al., 2024). As the Bayesian model selection framework already downweights extreme parameter combinations (e.g., high porosity and large grain size) through the chosen independent prior distributions, and because the minimum misfit and most probable parameters are generally consistent with, e.g., the porosity-grain size relationship described in the literature (e.g., Díaz-Curiel et al., 2024), we do not expect a significant change in the posterior probabilities.

[Regarding ut, I respect the uncertainty that leads the authors to use a log-uniform prior, but as I recall, the Zoet-Iverson slip law includes a prediction for ut based on sediment properties (most notably grain size) which already has a relatively narrow range in this study. Given that, it doesn't seem reasonable to expect ut values near $10^4$ m/yr as equally likely as, say $10^2$? There are also at least two other studies I can recall that provide calculated values of ut in different configurations: Helanow et al. (2020; DOI: 10.1126/sciadv.abe7798) for sliding over rough, rigid beds and Hansen et al. (2024; DOI/10.1029/2023GL107681) for frozen sediments over till. Some discussion of this would be helpful, as it's not clear whether the wide prior range used here is physically justified.]

Zoet and Iverson (2020) report $u_{t,noN}$ values in the range 56.36 to 363.52 MPa$^{-1}$ m yr$^{-1}$. Because Hansen et al. (2024) use the same bed material (Horicon till sourced from same location) but with plowing clasts removed, they use the model parameters given in Table S1 in Zoet and Iverson (2020) except for a smaller clast radius $R = 0.0045$ m (instead of $R = 0.015$ m or $R = 0.030$ m), leading to $u_{t,noN} = 1120.17$ MPa$^{-1}$ m yr$^{-1}$. Given these significant uncertainties and that $u_{t,noN}$ depends on several other uncertain parameters, we argue that $u_{t,noN}$ is best represented by a log-uniform prior (currently covering the range 3.16 to 3155.76 MPa$^{-1}$ m yr$^{-1}$). Note that the regularised Coulomb law used in Helanow et al. (2021) is not the same as in Zoet and Iverson (2020). We included these additional details in the revised manuscript: The transition speed coefficient ($C_{ZI}$) values reported in the initial publication of the

Zoet-Iverson sliding law range from 56.36 to 363.52 MPa$^{-1}$ m yr$^{-1}$ (Zoet and Iverson, 2020). A later study using the same bed material (Horicon till sourced from the same location) but with plowing clasts removed uses the same parameters (given in Table S1 of Zoet and Iverson, 2020) except for a smaller clast radius $R = 0.0045$ m (instead of $R = [0.015, 0.030]$ m), leading to $C_{ZI} = 1120.17$ MPa$^{-1}$ m yr$^{-1}$ (Fig. S4 in Hansen et al., 2024). Given these significant uncertainties and that $C_{ZI}$ depends on several other uncertain parameters, a log-uniform prior covering the range 3.16 to 3155.76 MPa$^{-1}$ m yr$^{-1}$ was chosen (Fig. 3c).

[It would be helpful to emphasize more clearly in the introduction or discussion that the method presented here is primarily applicable to soft-bedded glacier systems, since the acoustic impedance contrast relies on wave propagation through a granular medium. This is an important distinction, especially considering that some of the tested sliding laws were originally formulated for rigid or mixed bed topographies. I think an open question remains in glaciology regarding how these different sliding laws apply across regions with spatially heterogeneous basal conditions (e.g., Maier et al., 2021, https://doi.org/10.5194/tc-15-1435-2021). The result that a fast-flowing, soft-bedded glacier like Pine Island Glacier exhibits Coulomb-style sliding is not surprising to me, given the preponderance of experimental and field evidence in the literature. But in light of continued and recent discussion in the literature (e.g., Law et al., 2024, https://doi.org/10.48550/arXiv.2407.13577) it would be worth emphasizing the both the utility and the limitation of the geophysical datasets to constrain the slip law.]

The referee is correct that, strictly speaking, the Viscous Grain-Shearing theory only applies to granular material. However, as outlined in detail in our response to the second referee, whenever we are using a sliding law originally formulated for hard beds (e.g., Budd, Schoof), we assume a granular, relatively undeformable material that cannot support tangential friction at its interface with the ice (here referred to as *rigid bed*). We added Strictly speaking, the VGS theory used to predict acoustic impedance only applies to granular material (Sec. 2.4). However, while the formation of cavities, for example, is most appropriate for undeformable bed protrusions, larger rock fragments embedded in granular sediment or even fine-grained deformable sediment might play a similar role (Schoof, 2007a,b; Fowler, 2009; Schoof et al., 2012). Therefore, whenever we are using a sliding law initially developed for hard bedrock (Sec. 2.2.3 and 2.2.6), we assume a granular, relatively undeformable material that can not support tangential friction at its interface with the ice (here referred to as *rigid bed*).

We agree with the referee that spatially heterogeneous basal conditions remain an open research question and spatially variable parameters (grain size, porosity, as well as sliding law parameters) should thus be explored in future studies. We added [...] incorporating spatially variable model parameters [...] should be explored in future studies.

**2 Author's response to Anonymous Referee 2 Comment 1**

**General comments**

We thank the referee for their constructive comments. A point-by-point reply is reported below, with referee comments in orange, our replies in black, and the revisions in light blue. We agree with the specific referee comments not listed here and have revised the manuscript accordingly. Specific comments that merely repeat points already addressed in the referee's general comments are also not listed here. References to figures, tables, and sections in our replies (black) refer to the original manuscript, whereas those in the revisions (light blue) correspond to the revised version.

[Bayesian approaches are generally used to determine the posterior probability density function (PDF) of model parameters, given prior information and constraining observations. For each sliding law, you thus obtain a posterior PDF as a function of the three chosen varying parameters. To obtain a probability for a sliding law, you integrate the posterior PDF over the three-dimensional parameter space (if I understand correctly). This last step is not justified at all in the manuscript while it is critical as all conclusion are based on this. It is not clear to me that a higher integrated probability over the whole parameter space makes a sliding law more likely than another. For example, a model with high but localized maximum PDF can have a lower score than smaller maximum PDF spread on a larger domain of the parameter space. The way the sliding law probability is calculated clearly needs theoretical background. This is critical for the paper as the data does not bring significant difference in misfit and thus data-based likelihood.]

The referee is correct to point out that Bayesian approaches are generally used to determine the posterior probability density function (PDF) of model parameters, given prior information and constraining observations. The situation that we consider here is slightly different, however, and is more akin to Bayesian model selection than the routine application of Bayes' rule for a single model. The main difference for the model selection framework is that the probability space is extended to cover multiple models, each of which has its own parameter space. Apart from that distinction, standard manipulations of probability are used, including Bayes' rule, marginalisation, and normalisation (e.g., Jaynes, 2003).

It is true that we do not justify these standard manipulations, but in the revised manuscript, we emphasise that nothing unusual is happening beyond a straightforward extension of the probability space to acknowledge the possibility of multiple different models.

The statement $\int_{\Theta_i} P(\Theta_i|M_i)\,d\Theta_i = 1$ says that once a model has been chosen, the parameters of that model must lie somewhere in its parameter space with certainty. This is self-evident. By performing this normalisation for each model, we take advantage of the well-known capacity of Bayesian model selection to automatically apply Occam's Razor. Overly flexible models with a large range or dimension of parameter space are penalised relative to simpler, less flexible models with fewer parameters or tighter bounds upon parameters.

The referee questions the marginalisation over the model parameters $\Theta_i$ (as expressed by equation 16), but this is standard because we wish to compare the posterior probabilies of models $P(M_i|D)$, not the joint posterior probability of models and parameters $P(\Theta_i, M_i|D)$. Note that $P(\Theta_i|M_i)$ is a conditional probability like any other and can be manipulated using standard rules of probability (e.g., Equation 16).

Unlike the more standard application of Bayes' rule, each model in the Bayesian model selection framework has its own particular parameter space, and this parameter space can be of any dimension. The two fixed effective pressure endmember scenarios have a 2D parameter space (grain size and porosity). All other sliding laws have a 3D or 4D parameter space (one or two additional sliding law parameters). The number of values examined for these additional sliding law parameters varies across different sliding laws. Having an additional dimension or a larger number of values examined effectively increases the chance of obtaining a good fit to the data, and this is compensated for appropriately in the Bayesian approach. The key idea is that a balance between goodness of fit and model flexibility is desirable, but we emphasise that no special manipulations are required to enforce this balance in the Bayesian approach, as it emerges quite naturally. We added However, the situation here slightly differs from the routine application of Bayes' rule for inferring model parameters within a single model and is more akin to Bayesian model selection. The

main difference for the model selection framework is that the probability space is extended to cover multiple models, each of which has its own parameter space. Since the number of parameters differs between models (e.g., two for the fixed effective pressure scenarios and four for the Zoet-Iverson sliding law) and we aim to compare the posterior probabilities of models $P(M_i|D, I)$, not the joint posterior probability of models and parameters $P(\Theta_i, M_i|D, I)$, we marginalize over the model parameters $\Theta_i$ to retrieve $P(D, I|M_i)$: and This normalization reflects the fact that once a model has been chosen, the parameters of that model must lie somewhere within its parameter space with certainty. This is self-evident and automatically applies Occam's Razor, penalizing models with a larger parameter space compared to less flexible models. The key idea of Occam's Razor is that a balance between goodness of fit and model flexibility is desirable, but we emphasise that no special manipulations are required to enforce this balance in the Bayesian approach.

[I do not understand why you are limiting your parameter space to three varying parameters. I suspect this is because you do an exhaustive grid search to build the posterior PDF. There are simple methods such as the Monte Carlo algorithm, that can be used to efficiently calculate the posterior PDF in cases where the parameter space is large. This would be easy to implement in your case where the forward model is fast to compute. This limitation forces you to calculate two different probabilities for some sliding laws (Schoof and Zoet-Iverson), where you arbitrarily fix one of the sliding parameters. This makes no sense to me, especially when you assume $\mu = C_{\max} = 0.5$ without justification (when varying ut or $C_s$). The PDF should be built with varying all relevant parameters together.]
The referee is correct that we limited the parameter space to three dimensions due to the computational cost of the grid search. Therefore, our previous results more precisely identified which of the models with three or fewer dimensions best represent the measured acoustic impedance data. We have since expanded the parameter space to four dimensions for all sliding laws with previously two different three-dimensional representations (Tsai-Budd, Schoof, and Zoet-Iverson) and discuss these results in the revised manuscript (see track-changes file for details). We agree that methods to simultaneously explore even more parameters, e.g. different exponents, should be explored in future studies, and added Due to the computational cost of the grid search, we currently limit the model parameter space $\Theta_i$ to 4D. For example, we do not consider variations in the exponents $m$, $q$, and $p$ (Sec. 2.2). However, computationally more efficient methods, such as Monte Carlo algorithms, can be explored in future studies to simultaneously vary more than four parameters.

[I do not agree with the claim you are testing the Weertman law. You are simply testing the hypothesis of uniform effective pressure which as nothing to do with the Weertman law. If you want to say that the Weertman law is not appropriate you should show that the inverted $\tau_b$ as a function of $u_b$ does not match a power law. A figure showing the inverted $\tau_b$ as a function of $u_b$ is missing in the manuscript in any case.]
It is true that we can not directly test the Weertman sliding law or any sliding law, for that matter, that has no effective pressure dependence. To clarify this, we refrain from using the term *Weertman-type endmember scenarios* and instead refer to these experiments as fixed effective pressure endmember scenarios. However, the effective pressure is only uniform in the $N = 0$ Pa case, as all other fractions of the ice overburden pressure vary spatially due to the dependence on ice thickness. We added The most straightforward approach for estimating the effective pressure ($N$) – one that does not require the specification of a sliding law – is to assume it is at a fixed fraction of the ice overburden pressure ($p_i$) everywhere. To contextualize and constrain the results obtained using effective pressures derived from various sliding laws (Sec. 2.2.3 to 2.2.7), we compute the acoustic impedance corresponding to different fractions of the ice overburden pressure, including the two fixed effective pressure endmember scenarios; a lower bound $N = 0$ Pa for which the ice is assumed to be at floatation everywhere, and b) an upper bound, $N = p_i$, for which the effective pressure is assumed equal to the ice overburden pressure everywhere. These endmembers correspond, respectively, to situations where basal water pressure fully supports the weight of overlying ice or does not support any weight at all. and As Eq. 2 does not depend on the effective pressure, the Weertman-type power law can not be directly tested within this approach. Instead, we calculate the acoustic impedance for the Budd sliding law.

The relation between the inverted $\tau_b$ and $u_b$ is the same for all sliding laws and, therefore, provides by

itself no information on which sliding law is most appropriate. Therefore, we refrain from adding a figure showing the inverted $\tau_b$ as a function of $u_b$.

[I do not think you are able to distinguish which of the stress bounded sliding law perform better when the result is so dependent of the design of the Bayesian approach. You also hide that the Schoof law is almost the exact same law as the Zoet-Iverson law. You can indeed write the equation (7) of the manuscript in this form:]

$$\tau_b = C_{\max} N \left( \frac{u_b}{u_b + \left( \frac{C_{\max}}{C_s} N \right)^{\frac{1}{m}}} \right)^m \tag{1}$$

This is very similar to Zoet and Iverson with $p = 1/m$, $\mu = C_{\max}$ and $u_t = (C_{\max}/C_s N)^{1/m}$. The only difference is that $u_t$ is a function of $N^{1/m}$ in the Schoof formulation and a function of $N$ in Zoet-Iverson. We agree that it is difficult to select a single-best sliding law due to the small differences in posterior probabilities between some of the sliding laws incorporating a Coulomb friction term (Coulomb, Tsai-Budd, Schoof, Zoet-Iverson). For this reason, we focus on the distinction between the Coulomb-type and non-Coulomb-type sliding laws (fixed $N$ endmember scenarios and Budd). We added However, the Schoof and Zoet-Iverson sliding laws show a similarly strong increase, hindering the determination of a single-best sliding law.

Generally speaking, whenever we are using a sliding law originally formulated for hard beds (e.g., Budd, Schoof), we assume a granular, relatively undeformable material that cannot support tangential friction at its interface with the ice (here referred to as *rigid bed*). The formation of cavities, for example, is most appropriate for undeformable bed protrusions, but larger rock fragments embedded in granular sediment or even fine-grained deformable sediment might play a similar role (Schoof, 2007a,b; Fowler, 2009; Schoof et al., 2012). The basal drag for rigid beds is dominated by the deformation of ice around bed obstacles (form drag). In contrast, basal conditions dominated by skin drag are covered by the soft bed (deformable sediment) sliding laws (e.g., Coulomb, Zoet-Iverson).

While the form of the Schoof and Zoet-Iverson sliding law is indeed very similar, the physical reasoning and interpretation differ. As described above, the Schoof sliding law is most applicable for ice sliding over a rigid bed (granular but relatively undeformable material). It allows for the formation of cavities and incorporates Iken's bound ($C_{\max} = \tan \beta$; Iken, 1981; Schoof, 2005). $\beta$ is the maximum up-slope angle of the bed in flow direction. In contrast, the Zoet-Iverson sliding law aims to describe ice sliding over a water-saturated till bed (deformable). $\mu = \tan(\Phi)$ is the Coulomb friction coefficient and $\Phi$ the till friction angle. Thus, the two sliding laws represent different basal conditions, and $\mu$ and $C_{\max}$ describe different physical properties. We added While the mathematical form of the Schoof (Eq. 7) and Zoet-Iverson sliding law (Eq. 10) is very similar, the physical reasoning and interpretation differ. The Schoof sliding law is most applicable for ice sliding over a rigid bed (granular but relatively undeformable material), whereas the Zoet-Iverson sliding law aims to describe ice sliding over a water-saturated till bed (deformable). Similarly, the sliding-law-specific parameters $\mu$ and $C_{\max}$ represent distinct physical properties, and, may therefore differ significantly (Sec. 2.5).

[Posterior PDF are not shown, it would be usefull to have them in some figures to discuss the influence of prior PDF.]
We examine the influence of the prior distribution by applying (log-)uniform priors to all parameters (Fig. 6 vs. S22). As showing 2D planes of the 3D or 4D posterior PDF might be misleading, and to keep the manuscript concise (7 additional plots would be required), we refrain from adding additional map plots.

[Given the resolution of Bedmap-2, the estimation of $C_{\max}$ based on basal topography observation does not make any sense. Even if the inversion is performed at the kilometer scale, the relevant scale at which to estimate $C_{\max}$ is the meter scale, as this is the scale at which shear resistance is built. Also, the impedance model is based on the assumption of a sediment layer, which is inconsistent with the estimation of $C_{\max}$ based on the hard-bed theory. I do not see why $\mu$ and $C_{\max}$ should have different priors, given that they

play the same role in the friction law. Doing so favours one sliding law based on unjustified choices.]

While we agree that shear resistance is most likely built at scales smaller than the resolution of Bedmap-2, the bed roughness and therefore the actual relevant scale are less clear and likely vary spatially. However, these smaller scales will not be explicitly represented by the basal drag derived from the inversion. Therefore, the $C_{max}$ prior is determined by a combination of the Bedmap-2 bed angles and autonomous underwater vehicle data (2 m resolution), taking smaller resolutions into consideration. To clarify this, we moved parts of the description of the $C_{max}$ prior from the supplement to the main manuscript and added further details: While shear resistance is most likely built at scales smaller than the resolution of Bedmap-2, the bed roughness and therefore the actual relevant scale are less clear and likely vary spatially. As these smaller scales are not explicitly represented by the basal drag derived from our inversion, it is not straightforward to determine the $C_{max}$ prior directly from the small-scale AUV data. Therefore, we align the highest probability in the $C_{max}$ prior with the steepest Bedmap-2 bed angles and incorporate even steeper bed angles at smaller scales through a more gradual decline towards higher $C_{max}$ values (Sec. S6.2).

The referee is correct that, strictly speaking, the Viscous Grain-Shearing theory only applies to granular material. However, as outlined in detail above, this is consistent with our definition of rigid beds (granular but relatively undeformable material). Furthermore, glacier beds, e.g. the bed beneath Thwaites glacier, often do not support the clear differentiation between rigid beds and soft sediments assumed in the derivation of sliding laws. Instead, the bed might consist of a thin, deformable sediment layer draped over a rigid bed or alternating patches of sediment and rigid bed. Ultimately, the goal of this study is to find the basal sliding parameterisation that best captures the basal conditions identified by the acoustic impedance measurements. As the referee pointed out, by assuming $C_{max} = \mu$, we would effectively be testing two very similar sliding laws, which undermines this objective. Following this logic, and since $\mu$ and $C_{max}$ describe different physical properties, there is no reason why the two parameters should have the same prior.

[The title is a too strong statement compared to what you are actually able to infer. Furthermore you focus only on Pine Island glacier, not all Antarctica. I would propose instead: "Evidence of stress bounded friction law at Pine Island Glacier (Antarctica) inferred from seismic observations."]

We agree that the title is misleading and changed it to Inferring the ice sheet sliding law from seismic observations: A Pine Island Glacier case study. However, while we only infer the sliding law for Pine Island Glacier, the methodology developed here can be applied to acoustic impedance measurements from any glacial environment with granular material at the bed.

**Specific comments**

[L123 - By doing this you are not testing the Tsai law anymore .... This is the Budd part which make Tsai less likely in your result. I would remove the Tsai law as you cannot really test it.]

It is correct that we examine the Tsai-Budd instead of the Tsai sliding law itself and we clarified this in the revised manuscript: As for the Weertman-type power law itself, Eq. 5 can not be tested in the context discussed here because the Weertman part of the sliding law has no dependence on the effective pressure. To overcome this issue, we replace the Weertman part of Eq. 5 with the Budd sliding law (Eq. 3): However, due to its unique concept and mathematical form, the Tsai-Budd sliding law provides valuable insights, and we, therefore, prefer to keep it as part of the analysis.

[L125 - why this value ?]
and
[L137 - you should mention that you fixed Cmax=0.5 when varying Cs and explain why]
and
[Fig. 5 - why 0.5 ?? The best $\mu$ is 0.23. I expect you would use the best value found when varying $u_t$]

$\mu = 0.5$ is the Coulomb friction coefficient with the highest prior probability. Following this logic, we initially set $C_{max} = 0.2$ (the value with the highest prior probability). However, this value led to a high percentage of incompatible $u_b - \tau_b$ pairs and we, therefore, increased it to $C_{max} = 0.5$.

Using $\mu = 0.23$ would favour the Zoet-Iverson sliding law compared to the other sliding laws, as this

information only becomes available through running the experiments. For the Tsai-Budd sliding law when varying $\mu$, we previously relied on the referee's suggested approach as our prior knowledge about $C_\mathrm{B}$ is limited (log-uniform prior). However, all of the above is no longer an issue, since we now vary all four parameters simultaneously.

[L150 - you could call this parameter differently as it is not a speed anymore...something like "transition speed coefficient" and write it $1/C_\mathrm{zi}$. this would be more consistent with the Schoof law where a similar coefficient is equal to $(C_\mathrm{max}/C_s)$. So you would have in the schoof law: $u_t = (C_\mathrm{max}/C_s * N)^{1/m}$ and in the Zoet-Iverson law: $u_t = (1/C_\mathrm{zi})N$.]

We agree that transition speed coefficient is a better description of $u_\mathrm{t,noN}$ and revised the manuscript as follows: $u_\mathrm{t,noN} = C_\mathrm{ZI} = u_t/N$.

[L217 - Based on what the prior values are chosen?]

Following the suggestion of the first referee, a table containing detailed information on the porosity and grain size data as well as further details for the priors of the specific sliding law parameters (e.g., $C_\mathrm{max}$) were added to the revised manuscript (see Table S1 and track-changes file for details).

[Fig. 6 - why Schoof($C_s$) is not here ?]

Schoof($C_s$) was not included here because of the large number of incompatible $u_b$–$\tau_b$ pairs. We added further information regarding this issue in Sec. S5 of the revised supplement and by explicitly including the prior information from the inverted $u_\mathrm{b}$–$\tau_\mathrm{b}$ in the Bayesian equations (Sec. 2.5, see track-changes file for details).

[L266 - you should give the MAP parameters]

The MAP parameters are listed in Fig. S21 and S23. We added a reference to these figures here.

[L299 - this comes from the friction law, not directly the modeled impedance. It should be clear.]

The effective pressure is calculated using the friction law, but the friction law parameter used in this calculation is inferred from the acoustic impedance misfit. We added Since the predicted acoustic impedance depends on the effective pressure, an ice sheet sliding law and its parameters can be inferred, subsequently enabling the derivation of an effective pressure map.

**References**

Atapour, H. and Mortazavi, A.: The effect of grain size and cement content on index properties of weakly solidified artificial sandstones, Journal of Geophysics and Engineering, 15, 613, https://doi.org/10.1088/1742-2140/aaa14a, 2018.

Atre, S. R. and Bentley, C. R.: Laterally varying basal conditions beneath ice Streams B and C, West Antarctica, Journal of Glaciology, 39, 507–514, https://doi.org/10.3189/s0022143000016403, 1993.

Blankenship, D. D., Bentley, C. R., Rooney, S. T., and Alley, R. B.: Till beneath ice stream B. 1. Properties derived from seismic travel times, Journal of Geophysical Research, 92, 8903–8911, https://doi.org/10.1029/JB092iB09p08903, 1987.

Díaz-Curiel, J., Biosca, B., Arévalo-Lomas, L., Paredes-Palacios, D., and Miguel, M. J.: On the Influence of Grain Size Compared with Other Internal Factors Affecting the Permeability of Granular Porous Media: Redefining the Permeability Units, Lithosphere, 2024, lithosphere_2023_231, https://doi.org/10.2113/2024/lithosphere_2023_231, 2024.

Fowler, A. C.: Instability modelling of drumlin formation incorporating lee-side cavity growth, Proceedings of the Royal Society A: Mathematical, Physical and Engineering Sciences, 465, 2681–2702, https://doi.org/10.1098/rspa.2008.0490, 2009.

Gupta, A. and Ramanathan, A. L.: Grain texture as a proxy to understand porosity, permeability and density in Chandra Basin, India, SN Applied Sciences, 1, 1, https://doi.org/10.1007/s42452-018-0001-3, 2018.

Hansen, D. D., Warburton, K. L. P., Zoet, L. K., Meyer, C. R., Rempel, A. W., and Stubblefield, A. G.: Presence of Frozen Fringe Impacts Soft-Bedded Slip Relationship, Geophysical Research Letters, 51, e2023GL107681, https://doi.org/https://doi.org/10.1029/2023GL107681, e2023GL107681 2023GL107681, 2024.

Helanow, C., Iverson, N. R., Woodard, J. B., and Zoet, L. K.: A slip law for hard-bedded glaciers derived from observed bed topography, Science Advances, 7, 2–9, https://doi.org/10.1126/sciadv.abe7798, 2021.

Iken, A.: The Effect of the Subglacial Water Pressure on the Sliding Velocity of a Glacier in an Idealized Numerical Model, Journal of Glaciology, 27, 407–421, https://doi.org/10.3189/s0022143000011448, 1981.

Jaynes, E. T.: Probability Theory: The Logic of Science, Cambridge University Press, 2003.

Schoof, C.: The effect of cavitation on glacier sliding, Proceedings of the Royal Society A: Mathematical, Physical and Engineering Sciences, 461, 609–627, https://doi.org/10.1098/rspa.2004.1350, 2005.

Schoof, C.: Cavitation on Deformable Glacier Beds, SIAM Journal on Applied Mathematics, 67, 1633–1653, https://doi.org/10.1137/050646470, 2007a.

Schoof, C.: Pressure-dependent viscosity and interfacial instability in coupled ice–sediment flow, Journal of Fluid Mechanics, 570, 227–252, https://doi.org/10.1017/S0022112006002874, 2007b.

Schoof, C., Hewitt, I. J., and Werder, M. A.: Flotation and free surface flow in a model for subglacial drainage. Part 1. Distributed drainage, Journal of Fluid Mechanics, 702, 126–156, https://doi.org/10.1017/jfm.2012.165, 2012.

Wang, J.-P., François, B., and Lambert, P.: Equations for hydraulic conductivity estimation from particle size distribution: A dimensional analysis, Water Resources Research, 53, 8127–8134, https://doi.org/https://doi.org/10.1002/2017WR020888, 2017.

Zoet, L. K. and Iverson, N. R.: A slip law for glaciers on deformable beds, Science, 368, https://doi.org/10.1126/science.aaz1183, 2020.

---

## Author Response (AR2)

**Author's response to Referee Comments**

November 7, 2025

**1 Author's response to the Editors Comment**

[Thanks very much for your resubmission to The Cryosphere. As you'll see in reviews, Report #2 highlights a remaining issue with your estimation of prior probability for the Schoof C_max parameter – although the reviewer also notes that the overall message in the paper would be unchanged, even if it may impact the ranking of friction laws. I therefore wonder if a compromise could be reached here: is there any scope to at least consider the impact of a higher C_max parameter, as per the reviewer's suggestion? I'm not suggesting that models are run or revised in their entirety, but instead that a few lines of text are included to consider the impact of a different parameterisation.]

We thank the editor for their comment. We have revised the manuscript as outlined below; however, for the reasons discussed, we prefer to retain our original prior for the results presented in the main manuscript.

**2 Author's response to Anonymous Referee 2 Comment 2**

We thank the referee for their constructive comment. Our reply is reported below, with the referee comment in orange, our reply in black, and the revisions in light blue.

[However, there is still one important point on which I disagree. Since the mathematical formulations of the Schoof and Zoet-Iverson laws are similar, the ranking of these two laws is largely influenced by the choice of prior PDF for the model parameters. One could argue that this is justified because the parameters are associated with different physics and could thus have different priors, with which I agree. What I disagree with, is how the prior for C_max in the Schoof law is chosen based on irrelevant observations. Using different priors for $\mu$ and C_max favors the Zoet-Iverson law for an unjustified reason, in my opinion. This is evident in Figure S25, where the absence of a prior puts Coulomb and Schoof on a better score than Zoet-Iverson.]

[The claimed value of C_max = 0.2, as a maximum prior, is based on observations of bed topography that are not of a sufficiently high resolution to determine C_max. Furthermore, in three-dimensional geometries with a complete roughness spectrum, it is unclear how to relate bed slope and C_max because the relevant scale controlling friction is not known a priori and even associating a slope distribution with C_max is not straightforward. The authors answer that the one kilometer scale is relevant because of similar resolution of their independence data makes no sense to me. Even if the average friction at the kilometer scale is targeted, C_max can sill be controlled by smaller scale roughness and must be determined differently. An observational study such as that by Gimbert et al. (2019) suggests C_max = 0.4, which seems to me to be a more reliable estimation, even though it comes from a different glacier.]

[For all the above reasons and for a fair comparison between the different laws, I would suggest not providing any prior on $\mu$ and C_max. These parameters are simply not constrained well enough by external data to be associated with a prior PDF.]

We agree with the referee that it is not straightforward how to relate the bed slope and $C_{\max}$ given the

different scales in bed roughness. However, we want to emphasise that our $C_{max}$ prior does consider high-resolution observations from autonomous underwater vehicles (AUVs; horizontal resolutions of 1.5 m and 2 m). We also agree with the referee that the resolution of our model inversion is not really relevant here and have updated the description of the $C_{max}$ prior in the manuscript as follows: Due to the range of spatial scales in bed roughness that can affect basal drag, estimating $C_{max}$ from observations of bed topography is not straightforward. We therefore base our $C_{max}$ prior (Fig. 3d) on a combination of coarse-resolution bed topography beneath PIG retrieved from Bedmap2 data (Fig. S7 and S8; Fretwell et al., 2013), as well as high-resolution autonomous underwater vehicle (AUV) data collected downstream of Thwaites Glacier (1.5 m; Graham et al., 2022) and under the Thwaites Eastern Ice Shelf (2 m; Wåhlin, unpublished data; Fig. S9 and S10). Although shear resistance is most likely built at spatial scales smaller than the resolution of Bedmap2, these data provide a conservative lower bound on $C_{max}$ (Sec. S6.2).

Similarly, Sec. S6.2 in the supplement was updated to The distribution of the up-slope angles of the bed in flow direction ($\beta$) and the corresponding Iken's bound ($C_{max} = \tan\beta$; Fig. S7) is examined for the center part of Pine Island Glacier (PIG; magenta box in Fig. S8). As the horizontal grid resolution of Bedmap2 is 1 km (Fretwell et al., 2013), the maximum up-slope angle (and therefore $C_{max}$) on smaller scales might be significantly steeper than suggested by the distribution in Fig. S7. For example, autonomous underwater vehicle (AUV) data collected downstream of Thwaites Glacier (1.5 m horizontal resolution; Graham et al., 2022) and under the Thwaites Eastern Ice Shelf (2 m horizontal resolution; Wåhlin, unpublished data) indicate that the maximum $C_{max} > 0.7$ (largest value tested within this study; Fig. S9). As the bed roughness and therefore the actual relevant scale are unknown and likely vary spatially, the chosen $C_{max}$ prior incorporates the coarse resolution Bedmap2 data as a conservative lower bound and accounts for the higher bed angles observed at smaller scales through a more gradual decline towards higher values.

Furthermore, we updated the discussion on the effect of prior distributions. It now states Even when using log-uniform prior distributions for scaling coefficients and uniform priors for other parameters – thus making no use of the Bedmap2 or AUV data to constrain the $C_{max}$ prior – the sliding laws incorporating a Coulomb friction term still yield the highest probabilities, with the Coulomb and Schoof sliding law showing the greatest increase (26.3 % for both; Fig. S25).

Since our $C_{max}$ prior assigns a relatively high probability to the $C_{max} = 0.4$ suggested by Gimbert et al. (2021) and we show the results for (log-)uniform priors in the supplement, we refrain from adjusting the $C_{max}$ prior used for the results presented in the main manuscript (as suggested by the referee). Prior distributions are inevitably somewhat subjective, and different authors are likely to adopt distinct priors depending on the prior information available to them. Furthermore, adjusting the prior distribution after conducting the experiments, analysing the results, and receiving reviewer feedback would no longer represent a true prior in the Bayesian sense, as it would inevitably be informed by the results rather than remain independent of it.

**References**

Fretwell, P., Pritchard, H. D., Vaughan, D. G., Bamber, J. L., Barrand, N. E., Bell, R., Bianchi, C., Bingham, R. G., Blankenship, D. D., Casassa, G., Catania, G., Callens, D., Conway, H., Cook, A. J., Corr, H. F., Damaske, D., Damm, V., Ferraccioli, F., Forsberg, R., Fujita, S., Gim, Y., Gogineni, P., Griggs, J. A., Hindmarsh, R. C., Holmlund, P., Holt, J. W., Jacobel, R. W., Jenkins, A., Jokat, W., Jordan, T., King, E. C., Kohler, J., Krabill, W., Riger-Kusk, M., Langley, K. A., Leitchenkov, G., Leuschen, C., Luyendyk, B. P., Matsuoka, K., Mouginot, J., Nitsche, F. O., Nogi, Y., Nost, O. A., Popov, S. V., Rignot, E., Rippin, D. M., Rivera, A., Roberts, J., Ross, N., Siegert, M. J., Smith, A. M., Steinhage, D., Studinger, M., Sun, B., Tinto, B. K., Welch, B. C., Wilson, D., Young, D. A., Xiangbin, C., and Zirizzotti, A.: Bedmap2: Improved ice bed, surface and thickness datasets for Antarctica, Cryosphere, 7, 375–393, https://doi.org/10.5194/tc-7-375-2013, 2013.

Gimbert, F., Gilbert, A., Gagliardini, O., Vincent, C., and Moreau, L.: Do Existing Theories Explain Seasonal to Multi-Decadal Changes in Glacier Basal Sliding Speed?, Geophysical Research Letters, 48, 1–10, https://doi.org/10.1029/2021GL092858, 2021.

Graham, A. G., Wåhlin, A., Hogan, K. A., Nitsche, F. O., Heywood, K. J., Totten, R. L., Smith, J. A., Hillenbrand, C. D., Simkins, L. M., Anderson, J. B., Wellner, J. S., and Larter, R. D.: Rapid retreat of Thwaites Glacier in the pre-satellite era, Nature Geoscience, 15, 706–713, https://doi.org/10.1038/s41561-022-01019-9, 2022.